# ITSPACE: Monotone Gaussian Optimal Transport Updates

**Woojoo Na** [1]   **Jennifer Dy** [1]

## Abstract

Covariance matrices serve as compact descriptors of feature distributions in many machine-learning pipelines, including domain adaptation and Gaussian embeddings. Under a centered Gaussian approximation, the *unregularized* Wasserstein–2 optimal-transport (OT) discrepancy admits a closed form on covariances given by the Bures–Wasserstein (BW) objective on the symmetric positive definite (SPD) cone. We propose ITSPACE (Iterative Transport for Stable Proximal Alignment of Covariance Embeddings), a proximal majorization–minimization method that directly optimizes this exact BW objective through closed-form updates in a square-root factorization. In exact arithmetic, each iteration satisfies a sufficient-decrease inequality for the BW objective; under inexact polar computations, we provide an explicit certificate-gap bound controlling deviations from exact descent. The resulting iterations preserve PSD structure by construction and naturally support rank-restricted factors, making ITSPACE well-suited as a lightweight inner-loop primitive in settings where adaptation must be performed from unlabeled target batches under strict step and compute budgets. Across real-world covariance-alignment benchmarks, ITSPACE reaches low-BW-gap solutions substantially faster than BW-gradient descent, methods based on other covariance geometries, and entropically regularized sample-OT baselines.

## 1. Introduction

Many machine-learning pipelines adapt feature distributions rather than raw inputs directly. In unsupervised domain adaptation, for example, distribution shift is often reduced by matching first- and second-order feature statistics between a source domain and a target domain. Correlation alignment (CORAL) and related methods use covariance alignment as a practical surrogate for aligning feature distributions (Sun & Saenko, 2016). Similar covariance operations also appear in normalization layers, whitening–recoloring transforms, metric learning modules, and recent Bures–Wasserstein normalization methods for SPD features (Wang et al., 2025). In probabilistic representation learning and variational inference, Gaussian approximations are frequently used to represent local uncertainty or latent distributions (Lambert et al., 2022; Diao et al., 2023). Recent generative modeling methods also use Gaussian optimal-transport structure inside Wasserstein flow matching and Schrödinger-bridge formulations (Haviv et al., 2025; Bunne et al., 2023). Across these examples, the covariance matrix is not merely a statistic computed after training. It is often part of the computational state that must be updated, stored, and passed to another module.

This paper studies covariance alignment under the Bures–Wasserstein (BW) distance. The BW distance is the closed-form Wasserstein–2 distance between centered Gaussian distributions, written directly in terms of their covariance matrices (Gelbrich, 1990; Takatsu, 2011; Bhatia et al., 2019). Thus, when a feature distribution is represented by a Gaussian approximation, or when a method explicitly updates a covariance matrix, minimizing the BW distance is equivalent to minimizing the exact unregularized Gaussian optimal transport objective. This is different from entropic optimal transport on empirical samples, which solves a regularized sample-transport problem (Sinkhorn & Knopp, 1967; Cuturi, 2013; Peyré & Cuturi, 2019). We do not assume that raw datasets such as images or feature clouds are globally Gaussian. The setting considered here is more specific: the object being updated is a covariance matrix, or a low-rank factor representing that covariance.

At first glance, covariance alignment may seem trivial. If the target covariance $\Sigma_\star$ is available, why not simply replace the current covariance by $\Sigma_\star$? This is not the setting faced by many adaptation and representation-learning pipelines. During training, test-time adaptation, or minibatch-based domain correction, the model may maintain only a compact covariance state rather than an arbitrary dense $d \times d$ matrix.

[1]Department of Computer Engineering, Northeastern University, Boston, USA. Correspondence to: Woojoo Na <na.w@northeastern.edu>.

*Proceedings of the 43rd International Conference on Machine Learning*, Seoul, South Korea. PMLR 306, 2026.

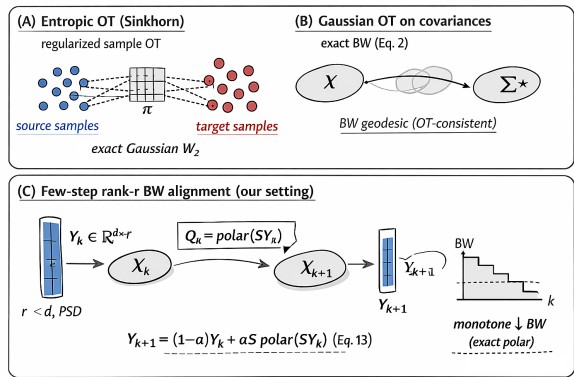

*Figure 1.* **Entropic OT vs. Gaussian OT on covariances.** Entropic OT solves a regularized sample-transport problem on empirical measures. In contrast, centered Gaussian OT has a closed-form Bures–Wasserstein objective on covariance matrices, and its geodesics are exact Gaussian displacement interpolations. This work focuses on updating low-rank covariance factors under this exact Gaussian OT objective.

A common representation is a low-rank factorization

$$X = YY^\top, \qquad Y \in \mathbb{R}^{d \times r}, \qquad r \ll d,$$

which guarantees that $X$ is positive semidefinite and has rank at most $r$. Such a factor is cheaper to store and update, and it can be inserted directly into downstream normalization, whitening, metric, or covariance-correction modules. The challenge is therefore not to write down the full target covariance once. The challenge is to produce valid low-rank covariance updates that move toward $\Sigma_\star$ under the BW objective after only a small number of steps.

This limited-update setting is important because many ML systems cannot afford to run a full matrix optimization routine inside every adaptation loop. Target statistics may change across domains, minibatches, or time, and the updated covariance may be consumed immediately by a downstream module. Consequently, intermediate iterates matter: after one or two updates, the covariance should already be positive semidefinite, rank constrained, and closer to the target under the exact Gaussian Wasserstein objective. Post-hoc projection is not an ideal solution, because it can add substantial cost and may break the connection between the update step and the objective being optimized.

Several existing methods address related problems, but they do not simultaneously provide a cheap low-rank factor update and descent under the exact BW objective. The BW geodesic gives the exact Gaussian optimal transport displacement interpolation between two Gaussian measures (Takatsu, 2011; Bhatia et al., 2019). However, it is an interpolation between covariance matrices and does not directly provide an iterative rank-constrained factor update. Gradient-based and Riemannian methods can optimize the

BW objective on SPD matrices (Han et al., 2021; Fan et al., 2024), but they typically require repeated dense matrix square roots, inverses, logarithms, or projections. Euclidean, log-Euclidean, and affine-invariant Riemannian updates are useful geometries for positive definite matrices, but they optimize different objectives. Entropic OT methods such as Sinkhorn are powerful for empirical measures, but their regularized objective is not the closed-form BW distance between Gaussian covariances.

We propose ITSPACE (Iterative Transport for Stable Proximal Alignment of Covariance Embeddings), a majorization–minimization method for rank-constrained covariance alignment under the BW distance. The method works directly with the low-rank factor $Y$. Using the identity

$$W_2^2(YY^\top, \Sigma_\star) = \|Y\|_F^2 + \mathrm{tr}(\Sigma_\star) - 2\|\Sigma_\star^{1/2} Y\|_*,$$

the BW objective becomes a quadratic term minus a nuclear norm. At each iteration, ITSPACE replaces the nuclear norm by a tight linear certificate obtained from a polar/Procrustes problem, adds a proximal term, and minimizes the resulting quadratic upper bound in closed form. The resulting update moves the current factor toward an aligned square root of the target covariance, while every covariance iterate remains positive semidefinite and rank constrained by construction.

Our contributions are as follows.

- **A low-rank update for BW covariance alignment.** We derive a closed-form factor update for aligning a covariance $X = YY^\top$ to a target covariance $\Sigma_\star$ under the exact Gaussian Wasserstein objective. Every iterate remains positive semidefinite and rank constrained by construction.

- **Descent guarantees for the true BW objective.** We prove monotone decrease of the BW objective under exact polar computation, strengthen this guarantee with a sufficient-decrease bound, and quantify the effect of approximate polar computations. We also include a matched-rank positive semidefinite extension in the appendix, together with the certificate condition needed in singular cases.

- **Empirical evaluation under limited update budgets.** Across real feature-covariance alignment tasks and downstream covariance-drift correction tasks, ITSPACE reaches BW alignment targets substantially faster than BW gradient descent and achieves competitive downstream recovery under the same rank constraint. In our downstream drift-correction experiments, one or two unlabeled target-batch updates recover a substantial fraction of the drift-induced degradation, including gains of $+3.8$ AUROC on Camelyon17 and $+5.1$ accuracy on Terra.

## 2. Background and Related Work

### 2.1. Sample OT vs. Gaussian OT

Optimal transport can be applied at different levels of representation. In sample-based OT, the inputs are empirical measures supported on data points. In Gaussian OT, the inputs are Gaussian distributions, so the transport problem can be written directly in terms of their means and covariances. This paper studies the latter setting: the object being updated is a covariance matrix, or a low-rank factor representing that covariance.

Given probability measures $\mu, \nu$ on $\mathbb{R}^d$ with finite second moments, the squared Wasserstein–2 distance is the optimal value of the Kantorovich problem

$$W_2^2(\mu, \nu) \; := \; \min_{\pi \in \Pi(\mu,\nu)} \int \|x - y\|_2^2 \, d\pi(x,y), \quad (1)$$

where $\Pi(\mu,\nu)$ denotes the set of couplings with marginals $\mu$ and $\nu$ (Peyré & Cuturi, 2019; Villani, 2009). Throughout this paper, $W_2$ denotes this unregularized objective.

Entropic OT modifies (1) by adding an entropy or KL penalty to the coupling, leading to Sinkhorn iterations (Sinkhorn & Knopp, 1967; Cuturi, 2013; Genevay et al., 2016; Feydy et al., 2019). This modification is crucial for scalable sample-based OT, but it changes the objective. Sinkhorn therefore solves a regularized empirical transport problem, whereas our target is the exact unregularized Gaussian Wasserstein distance available in closed form on covariances.

For centered Gaussian distributions, the unregularized $W_2$ distance depends only on covariance matrices. Let $\mathbb{S}_{++}^d$ denote the cone of $d \times d$ real symmetric positive definite matrices. If $X, \Sigma \in \mathbb{S}_{++}^d$, then

$$W_2^2(X, \Sigma) \; = \; \operatorname{tr}(X) + \operatorname{tr}(\Sigma) \; - \; 2 \operatorname{tr}\left((\Sigma^{1/2} X \Sigma^{1/2})^{1/2}\right), \quad (2)$$

where we write $W_2^2(X, \Sigma)$ for $W_2^2(\mathcal{N}(0, X), \mathcal{N}(0, \Sigma))$ (Dowson & Landau, 1982; Gelbrich, 1990; Takatsu, 2011; Bhatia et al., 2019). This is the Gaussian Bures–Wasserstein (BW) objective on covariances. Although (2) is often stated for positive definite covariances, it extends continuously to positive semidefinite covariances. This extension is essential for the low-rank representation $X = YY^\top$ used below, since such covariances are generally positive semidefinite rather than positive definite.

For non-Gaussian distributions, the covariance expression in (2) should not be interpreted as the full distributional $W_2$ distance. Rather, together with mean terms, it appears in Gelbrich-type lower bounds on $W_2$ and is used here as a second-order description of distributional discrepancy (Gelbrich, 1990).

### 2.2. BW vs. Other Geometries on Covariances

Covariance alignment is widely used to reduce mismatch between feature distributions. CORAL and related domain-adaptation methods match second-order feature statistics across source and target domains (Sun & Saenko, 2016). Covariance matrices also appear in whitening and recoloring transforms, normalization layers, metric-learning modules, and methods for SPD-valued features (Wang et al., 2025). These settings motivate algorithms that update covariances while respecting positive semidefinite structure.

There are several natural geometries on covariance matrices. Common choices include the Euclidean or Frobenius geometry, the log-Euclidean geometry, the affine-invariant Riemannian geometry, and the Bures–Wasserstein geometry (Moakher, 2005; Arsigny et al., 2006; Bhatia, 2007; Bhatia et al., 2019). These geometries are all useful, but they define different objectives. A step that decreases a Euclidean, log-Euclidean, or affine-invariant distance between covariances does not necessarily decrease the Gaussian Wasserstein objective in (2). Since our goal is Gaussian OT on covariances, BW is the objective optimized and reported in our main alignment experiments.

This objective choice also clarifies how to interpret baselines. Methods based on other SPD geometries are meaningful covariance-alignment baselines, but they are not descent methods for the BW objective. Sinkhorn methods are meaningful sample-OT baselines, but they solve a regularized empirical transport problem. Our comparisons therefore distinguish between the target objective, the matrix representation, and the computational constraints imposed by low-rank covariance updates.

### 2.3. Full-Covariance Paths vs. Low-Rank Factor Updates

Within Gaussian OT, a natural baseline is the BW geodesic. BW geodesics are the exact Wasserstein displacement interpolations between Gaussian measures (Takatsu, 2011; Bhatia et al., 2019). They provide a closed-form path between two covariance matrices and are therefore the Gaussian analogue of OT displacement interpolation.

However, the BW geodesic is a full-covariance path. It does not directly solve the update problem considered in this paper, where the maintained state is a low-rank factor

$$X = YY^\top, \qquad Y \in \mathbb{R}^{d \times r}, \qquad r \ll d.$$

A geodesic step can be followed by rank truncation, but that introduces an extra projection and does not give a closed-form descent step for the rank-constrained factor objective.

Gradient-based and Riemannian methods can also optimize the BW objective on the SPD cone (Han et al., 2021; Fan et al., 2024). These methods target the correct Gaus-

sian Wasserstein distance, but they typically involve repeated dense matrix square roots, inverses, logarithms, linear solves, or projections when a rank constraint is imposed. They may also require step-size tuning or line search to obtain reliable descent.

The gap addressed by ITSPACE is a low-rank update problem: we seek a closed-form update for the factor $Y$ itself, so that every iterate $X_k = Y_k Y_k^\top$ is positive semidefinite, rank constrained, and guaranteed to decrease the exact BW objective. The next section derives this update by rewriting the BW objective in the factor variable and minimizing a quadratic upper bound at each iteration. A detailed comparison of baseline families, objectives, guarantees, and per-step costs is provided in Appendix C.

## 3. The ITSPACE Algorithm

### 3.1. Problem Setup and Notation

Let $d \in \mathbb{N}$, and let $\mathbb{S}_{++}^d$ denote the cone of $d \times d$ symmetric positive definite matrices. We are given a fixed target covariance $\Sigma_\star \in \mathbb{S}_{++}^d$ and an initial covariance $X_0 \succeq 0$. The goal is to produce a short sequence of covariance iterates

$$X_0, X_1, \ldots, X_K$$

that moves the current covariance toward $\Sigma_\star$ under the exact Gaussian Wasserstein objective

$$W_2^2(X_k, \Sigma_\star) \qquad \text{as defined in Eq. (2).} \qquad (3)$$

The setting of interest is constrained by both a rank budget $r \ll d$ and a small iteration budget $K$. Thus the algorithm should maintain valid covariance iterates throughout the trajectory, rather than only after a final projection.

**Rank Budget and Factorization.** We enforce positive semidefiniteness and the rank constraint by representing the maintained state through a square-root factor:

$$X_k = Y_k Y_k^\top, \qquad Y_k \in \mathbb{R}^{d \times r}. \qquad (4)$$

This guarantees $X_k \succeq 0$ and $\operatorname{rank}(X_k) \leq r$ for every $k$. The current state of the algorithm is the factor $Y_k$; the corresponding covariance is $X_k = Y_k Y_k^\top$. When the available initial covariance is full-rank, we initialize the maintained rank-budgeted state by taking a rank-$r$ PSD truncation and a square-root factor of it. To keep notation simple, we write this maintained initial covariance as $X_0 = Y_0 Y_0^\top$.

The target covariance $\Sigma_\star$ may be full rank even when the maintained state is low rank. Our main theory assumes $\Sigma_\star \in \mathbb{S}_{++}^d$ so that its principal square root is well defined. In experiments with near-singular empirical covariances, we apply the same small diagonal stabilization across methods before computing matrix square roots (Appendix D.1).

The factorization is not unique: $Y$ and $YR$ represent the same covariance for any orthogonal matrix $R \in O(r)$. The update below chooses a convenient orientation of the factor through a polar/Procrustes certificate and then takes a closed-form proximal majorization–minimization step.

### 3.2. Deriving the Closed-Form Factor Update

We now derive ITSPACE as a proximal majorization–minimization (MM) method for minimizing the exact BW objective restricted to the rank-budgeted factorization $X = YY^\top$. Here $Y \in \mathbb{R}^{d \times r}$ denotes a generic factor variable and $Y_k$ denotes the current factor iterate.

**Factor Form of the BW Objective.** Let

$$S := \Sigma_\star^{1/2}$$

be the principal square root of the target covariance. Substituting $X = YY^\top$ into the BW closed form and using $\operatorname{tr}((AA^\top)^{1/2}) = \|A\|_*$ gives the equivalent factor objective

$$F(Y) := W_2^2(YY^\top, \Sigma_\star) = \|Y\|_F^2 + \operatorname{tr}(\Sigma_\star) - 2\|SY\|_*. \qquad (5)$$

This identity is the main reduction. The rank constraint is built into the factor $Y$, and the only nonquadratic term in $F$ is the nuclear norm $\|SY\|_*$. Since this nuclear norm appears with a negative sign, a linear lower bound on $\|SY\|_*$ becomes a quadratic upper bound on the objective. This is the basis of the MM update.

**Nuclear-Norm Certificate.** The nuclear norm admits the support-function representation

$$\|A\|_* = \max_{\|Q\|_2 \leq 1} \operatorname{tr}(Q^\top A), \qquad A \in \mathbb{R}^{d \times r}. \qquad (6)$$

For $A \in \mathbb{R}^{d \times r}$, we write $\operatorname{polar}(A)$ for a selected maximizer in (6). If $A = U\Gamma V^\top$ is a compact singular value decomposition, then one valid choice is

$$\operatorname{polar}(A) = UV^\top.$$

When $A$ has full column rank, this coincides with the usual thin polar factor $A(A^\top A)^{-1/2}$. When $A$ is rank deficient, the maximizer may not be unique and need not satisfy $Q^\top Q = I_r$; for example, when $A = 0$, the choice $Q = 0$ is feasible and attains the maximum.

At the current iterate, define

$$Q_k := \operatorname{polar}(SY_k), \qquad B_k := SQ_k. \qquad (7)$$

Then $\|Q_k\|_2 \leq 1$, and by (6), for every factor $Y$,

$$\|SY\|_* \geq \operatorname{tr}(Q_k^\top SY). \qquad (8)$$

If $Q_k$ is an exact maximizer for $SY_k$, then the inequality is tight at $Y = Y_k$. Intuitively, $Q_k$ selects an orientation that best aligns the current factor with the target square root, and $B_k = SQ_k$ is the factor used by the quadratic update below.

**Quadratic Upper Bound.** Because the nuclear norm appears with a negative sign in (5), the lower bound (8) gives an upper bound on $F$. For every $Y$,

$$
\begin{aligned}
F(Y) &= \|Y\|_F^2 + \mathrm{tr}(\Sigma_\star) - 2\|SY\|_* \\
&\leq \|Y\|_F^2 + \mathrm{tr}(\Sigma_\star) - 2\,\mathrm{tr}(Q_k^\top SY) \\
&= \|Y - B_k\|_F^2 + C_k,
\end{aligned} \tag{9}
$$

where

$$
C_k := \mathrm{tr}(\Sigma_\star) - \|B_k\|_F^2
$$

is independent of $Y$. Under an exact polar certificate, this quadratic majorizer is tight at the current iterate $Y_k$.

**Proximal Stabilization and Closed-Form Update.** We add a proximal term around $Y_k$ and minimize the resulting quadratic surrogate:

$$
U_k(Y) := \|Y - B_k\|_F^2 + \frac{1}{2\lambda}\|Y - Y_k\|_F^2, \qquad \lambda > 0. \tag{10}
$$

This surrogate has a closed-form minimizer. Writing

$$
\alpha := \frac{2\lambda}{1 + 2\lambda} \in (0,1), \tag{11}
$$

the update is

$$
Y_{k+1} = \alpha B_k + (1-\alpha)Y_k = \alpha S\,\mathrm{polar}(SY_k) + (1-\alpha)Y_k. \tag{12}
$$

The covariance iterate is then $X_{k+1} = Y_{k+1}Y_{k+1}^\top$. Thus each iteration consists of computing the polar certificate $Q_k = \mathrm{polar}(SY_k)$ and taking a closed-form average between the current factor $Y_k$ and the certificate-induced factor $B_k = SQ_k$.

**Role of $\lambda$ and $\alpha$.** The parameter $\alpha$ is not an independently chosen learning rate; it is the averaging weight induced by the proximal penalty. Large $\lambda$ gives a more aggressive step with $\alpha$ close to 1, while small $\lambda$ keeps the update close to the current factor. In the full-rank case $r = d$, if $Y_k$ is full rank, the undamped limit $\alpha \to 1$ maps $Y_k$ to a square root of $\Sigma_\star$ up to an orthogonal rotation. We use $\alpha < 1$ when a controlled multi-step trajectory is desired or when damping is helpful in finite precision.

### 3.3. Algorithm

Algorithm 1 summarizes the iteration for a single target covariance. The target square root $S = \Sigma_\star^{1/2}$ is computed once and then reused across iterations. Implementation details for computing $\mathrm{polar}(\cdot)$ efficiently and stably are provided in Appendix B.2.

---

**Algorithm 1** ITSPACE: Proximal MM for BW Covariance Alignment

---

**Require:** Target $\Sigma_\star \in \mathbb{S}_{++}^d$, initial factor $Y_0 \in \mathbb{R}^{d \times r}$, proximal weight $\lambda > 0$, iterations $K$
**Ensure:** Factors $\{Y_k\}_{k=0}^K$ and, when needed, covariances
 $\quad X_k = Y_k Y_k^\top$
 $\quad S \leftarrow \Sigma_\star^{1/2}$
 $\quad \alpha \leftarrow 2\lambda/(1 + 2\lambda)$
 $\quad$ **for** $k = 0, 1, \ldots, K-1$ **do**
 $\qquad Q_k \leftarrow \mathrm{polar}(SY_k)$
 $\qquad Y_{k+1} \leftarrow \alpha\,SQ_k + (1-\alpha)\,Y_k$
 $\quad$ **end for**

---

### 3.4. Computational Cost

The computation separates into one-time preprocessing, repeated low-rank updates, and optional BW evaluations for logging. Computing $S = \Sigma_\star^{1/2}$ once costs $O(d^3)$ time and $O(d^2)$ memory, for example via eigendecomposition. This cost is paid once per target covariance and the same $S$ is reused across all ITSPACE iterations.

Each iteration first forms $SY_k$, computes the polar factor of the resulting $d \times r$ matrix, and then forms $SQ_k$. The two multiplications involving $S$ cost $O(d^2 r)$ time. The polar factor of a $d \times r$ matrix can be computed by a thin SVD in $O(dr^2)$ time, or by a Gram-based route in $O(dr^2 + r^3)$ time when $r \ll d$; see Appendix B.2. Therefore, the repeated per-iteration update cost is

$$
O(d^2 r + dr^2 + r^3).
$$

For $r \ll d$, the repeated cost is dominated by multiplications with the fixed target square root $S$, rather than by a full $d \times d$ matrix square root at each step. In the full-rank case $r = d$, the update cost reduces to $O(d^3)$.

Evaluating $W_2^2(X_k, \Sigma_\star)$ exactly for plots or tables requires a matrix square root and typically costs $O(d^3)$ per evaluation. This evaluation overhead is not part of the ITSPACE update itself. In our timing protocol (Section 5.1), the one-time computation of $S$ is treated as preprocessing and excluded from per-iterate algorithmic time, so the reported algorithmic time reflects only the repeated inner-loop updates.

## 4. Theoretical Properties

We state the main guarantees for the update in Eq. (12). Throughout this section, the target covariance satisfies $\Sigma_\star \in \mathbb{S}_{++}^d$, and we write $S = \Sigma_\star^{1/2}$. For a factor $Y \in \mathbb{R}^{d \times r}$, define

$$
F(Y) := W_2^2(YY^\top, \Sigma_\star) = \|Y\|_F^2 + \mathrm{tr}(\Sigma_\star) - 2\|SY\|_*. \tag{13}
$$

Let $\{Y_k\}_{k=0}^K$ be generated by ITSPACE, and define $X_k := Y_k Y_k^\top$. We also use the notation

$$Q_k := \mathrm{polar}(SY_k), \qquad B_k := SQ_k, \qquad \alpha := \frac{2\lambda}{1 + 2\lambda}.$$

Here $\mathrm{polar}(\cdot)$ is understood in the support-function sense defined in Section 3.2. All proofs are deferred to Appendix A.

### 4.1. Validity and Sufficient Descent

The factorization immediately enforces the structural constraint required by the algorithm.

**Proposition 4.1** (Validity of the Iterates). *For every iteration $k$, the covariance iterate $X_k = Y_k Y_k^\top$ is positive semidefinite and satisfies $\mathrm{rank}(X_k) \leq r$. If $r = d$ and $Y_k$ is invertible, then $X_k \in \mathbb{S}_{++}^d$.*

The next theorem strengthens the monotonicity guarantee. The update does not only decrease the exact BW objective; it decreases it by at least a squared step-size term in factor space.

**Theorem 4.2** (Sufficient BW Descent). *Assume that $Q_k = \mathrm{polar}(SY_k)$ is an exact maximizer of the nuclear-norm support problem for $SY_k$. Let $Y_{k+1}$ be the ITSPACE update in Eq. (12). Then*

$$F(Y_k) - F(Y_{k+1}) \;\geq\; \frac{1}{\alpha}\|Y_{k+1} - Y_k\|_F^2 \;=\; \alpha\|B_k - Y_k\|_F^2. \tag{14}$$

*Consequently,*

$$W_2^2(X_{k+1}, \Sigma_\star) \leq W_2^2(X_k, \Sigma_\star) \qquad \textit{for all } k.$$

This result is the main descent certificate for ITSPACE. It applies for any factor rank $r \leq d$ under the positive definite target assumption above. A direct finite-budget consequence is

$$\sum_{k=0}^{K-1} \|Y_{k+1} - Y_k\|_F^2 \;\leq\; \alpha\big(F(Y_0) - F(Y_K)\big) \;\leq\; \alpha F(Y_0). \tag{15}$$

Thus the same certificate that proves monotonicity also controls the cumulative movement of the factor iterates. This is an objective-level guarantee for the exact BW energy; downstream task metrics need not improve monotonically because they also depend on the feature representation, the fixed predictor, and the shared downstream adaptation map. The proof follows from the tight quadratic upper bound induced by the polar certificate and a completion-of-squares identity for the proximal surrogate.

### 4.2. Inexact Polar Certificates

Theorem 4.2 assumes that the polar certificate is computed exactly. In finite precision, it is useful to state the effect of an inexact but feasible certificate.

**Proposition 4.3** (Inexact Polar Certificates). *Let $A_k := SY_k$, and let $\widehat{Q}_k$ satisfy $\|\widehat{Q}_k\|_2 \leq 1$ and*

$$\mathrm{tr}(\widehat{Q}_k^\top A_k) \;\geq\; \|A_k\|_* - \varepsilon_k \qquad \textit{for some } \varepsilon_k \geq 0. \tag{16}$$

*Form the surrogate by replacing $Q_k$ with $\widehat{Q}_k$, and let $Y_{k+1}$ be the corresponding minimizer. Then*

$$F(Y_{k+1}) \;\leq\; F(Y_k) + 2\varepsilon_k. \tag{17}$$

Thus finite-precision error enters the descent statement only through the certificate gap $\varepsilon_k$. When $\varepsilon_k = 0$, the monotonicity part of Theorem 4.2 is recovered. A sharper descent-margin version of Proposition 4.3, which recovers the full sufficient decrease bound in the exact case, is given in Appendix A.4.

### 4.3. Full-Rank Invertible Fixed Points

The factorization $X = YY^\top$ is invariant under right multiplication by an orthogonal matrix. The following proposition characterizes fixed points in the full-rank invertible case.

**Proposition 4.4** (Full-Rank Invertible Fixed Points). *Assume $r = d$ and $\Sigma_\star \in \mathbb{S}_{++}^d$. Restrict attention to invertible factor iterates, so that the polar factor of $SY$ is unique and orthogonal. Then the fixed points of the factor update are exactly*

$$Y = \Sigma_\star^{1/2} R, \qquad R \in O(d),$$

*and the corresponding covariance fixed point is uniquely*

$$X = YY^\top = \Sigma_\star.$$

The invertibility restriction is important. When factors or targets are singular, polar certificates can be nonunique, and a blanket fixed-point statement can fail under an unlucky certificate choice. Appendix A.7 gives a matched-rank PSD extension together with a counterexample showing why the certificate choice matters in singular cases. Additional equivariance properties under orthogonal changes of basis and joint rescaling are stated in Appendix A.6, with the necessary consistency condition for nonunique polar certificates.

## 5. Experiments

We evaluate ITSPACE as a few-step inner-loop optimizer for the Gaussian Bures–Wasserstein (BW) objective $W_2^2(\cdot, \Sigma_\star)$ (Eq. (2)). We study two settings: (i) *rank-budgeted covariance alignment*, where the goal is to reduce the exact BW objective under a fixed rank and step budget, and (ii) *CovDrift-MR downstream transfer*, where the aligned covariance is used inside a fixed prediction pipeline under controlled covariance drift. Unless stated otherwise, all alignment results are evaluated using the same closed-form BW value.

**Experiment Overview.** **Experiment I (optimization).** Given $X_0$ and $\Sigma_\star$, we run each method for $K=20$ steps under rank budget $r=16$ and report time-to-gap at $\tau \in \{0.1, 0.02\}$ using the algorithmic time axis $t_{\text{alg}}$ (update + rank projection; BW-evaluation excluded). **Experiment II (CovDrift-MR downstream).** We freeze a linear classifier trained on source features, inject a stationary rank-16 covariance drift into target features, and adapt using only unlabeled target batches. Under budgets $K \in \{1, 2, 5, 20\}$, each method outputs an aligned covariance and the corresponding whitening–recoloring transform, applied to target-test features before evaluation with the frozen head.

### 5.1. Metrics and Evaluation Protocol

Given $X_0$ and $\Sigma_\star$, each method produces iterates $\{X_k\}_{k=0}^K$, or factors $Y_k$ with $X_k = Y_k Y_k^\top$. We evaluate each iterate with the closed-form BW value $W_2^2(X_k, \Sigma_\star)$.

**Rank-Budgeted GAP and Time-to-GAP.** With $\text{rank}(X_k) \leq r < d$, the BW objective generally cannot reach zero. We therefore report a normalized gap above a method-independent rank-$r$ truncation reference. Let $\Pi_r(\cdot)$ be rank-$r$ PSD truncation and define

$$\text{floor}_r(\Sigma_\star) := \sum_{i=r+1}^{d} \lambda_i(\Sigma_\star), \qquad (18)$$

where $\lambda_i(\Sigma_\star)$ are eigenvalues in descending order. Since $\Pi_r(\Sigma_\star)$ shares eigenvectors with $\Sigma_\star$, $W_2^2(\Pi_r(\Sigma_\star), \Sigma_\star) = \text{floor}_r(\Sigma_\star)$. We normalize the remaining BW excess by

$$\text{gap}_r(X_k) := \frac{W_2^2(\Pi_r(X_k), \Sigma_\star) - \text{floor}_r(\Sigma_\star)}{W_2^2(\Pi_r(X_0), \Sigma_\star) - \text{floor}_r(\Sigma_\star)}. \quad (19)$$

Thus $\text{gap}_r(X_0) = 1$, and smaller values indicate closer alignment under the same rank budget. For methods that operate directly in rank-$r$ factors, $\Pi_r(X_k) = X_k$ and the projection is only notational. We report the first iterate with $\text{gap}_r(X_k) \leq \tau$ for $\tau \in \{0.1, 0.02\}$; if a threshold is not reached within $K$ steps, we report **NR**.

**Timing and Fairness.** We report cumulative *algorithmic time* $t_{\text{alg}} := t_{\text{update}} + t_{\text{proj}}$, where $t_{\text{proj}}$ is the explicit cost of enforcing the rank budget via truncation $\Pi_r(\cdot)$ for methods that produce full-rank iterates. Any method-internal computation required to produce an iterate, such as backtracking or line-search checks, is included in $t_{\text{update}}$. Shared BW-evaluation time for logging is excluded, and one-time preprocessing, such as forming $\Sigma_\star^{1/2}$, is excluded unless stated otherwise; both are reported in Appendix D.2. We report medians over seeds for timing and mean±std for downstream metrics.

### 5.2. Datasets

Each instance specifies a target covariance $\Sigma_\star$ and an initial covariance $X_0$. We use fixed representations and per-domain empirical covariances, aligning source covariances toward a designated target domain. The three alignment instances correspond to Camelyon17 hospital shift, VisDA-2017 synthetic-to-real shift, and Terra/CCT-20 location shift; details of feature extraction and covariance construction are in Appendix D.1.

**Rank and Iteration Budgets.** We use $r=16$ and $K=20$ throughout to reflect an inner-loop constraint in which the evaluated covariance state must remain rank constrained. Methods that produce full-rank iterates are projected to rank $r$ via $\Pi_r(\cdot)$ after each update, so all methods are compared under the same rank budget.

### 5.3. Experiment I: Covariance Alignment

Unless otherwise stated, we use $r=16$, $K=20$, and three seeds $\{0, 1, 2\}$, reporting medians. Figure 2 plots $\text{gap}_r$ versus $t_{\text{alg}}$ across all three covariance-shift instances. Across datasets, ITSPACE reaches the low-gap regime ($\text{gap}_r \leq 0.02$) earlier in $t_{\text{alg}}$ than BW-GD and methods based on other covariance geometries.

Table 1 reports time-to-gap across datasets; **NR** indicates that a threshold is not reached within $K$ steps, with parentheses showing $\text{gap}_r(X_K)$. Projection costs are included for full-rank baselines because enforcing the rank budget is part of the evaluated state constraint. Sinkhorn variants often fail to reach the low-gap threshold under BW evaluation within $K$ steps, which is expected because they optimize a regularized sample-OT objective rather than the exact Gaussian BW objective.

A speedup summary relative to BW-GD is reported in Appendix Table 6.

### 5.4. Experiment II: CovDrift-MR Downstream Transfer Using Aligned Covariances

We test whether faster few-step alignment improves downstream performance under a controlled covariance drift. For each dataset and seed, we train a fixed linear head on labeled source features, inject a rank-16 covariance drift into target features, and adapt using only unlabeled target batches. Each method produces an aligned covariance and the associated whitening–recoloring transform applied at test time. For Terra, we evaluate using a fixed closed-set protocol, using only target-test labels present in source training; labels are used only for evaluation and the filtering is identical across methods. All methods share the same feature standardization, moment estimates, stabilization, rank enforcement, and transport-map implementation; the

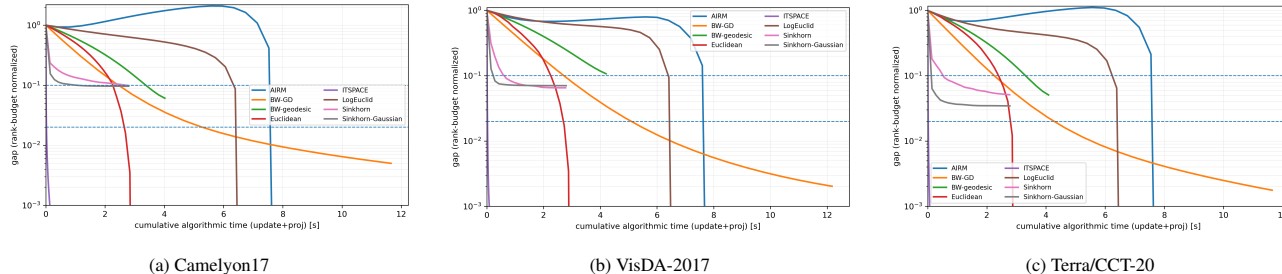

| (a) Camelyon17 | (b) VisDA-2017 | (c) Terra/CCT-20 |

*Figure 2.* **Rank-budget GAP contraction under exact BW evaluation.** All datasets use $d=2048$, $r=16$, $K=20$, and three seeds. Curves plot $\text{gap}_r$ versus $t_{\text{alg}}=t_{\text{update}}+t_{\text{proj}}$; dashed lines mark $\tau \in \{0.1, 0.02\}$. ITSPACE reaches the low-gap regime fastest under the shared rank-budget timing protocol.

*Table 1.* **Cross-dataset time-to-gap under a rank budget.** Median $t_{\text{alg}}$ (seconds) to reach $\text{gap}_r \leq 0.1$ / $\text{gap}_r \leq 0.02$ (Section 5.1). **NR** indicates not reached within $K=20$; parentheses report $\text{gap}_r(X_K)$. All settings use $r=16$ and three seeds $\{0, 1, 2\}$. Best (fastest) entries per dataset are in bold.

| Dataset | ITSPACE | BW-GD | Euclid. (+proj) | Log–Euclid. (+proj) | AIRM (+proj) | BW geodesic | Sinkhorn | Sinkhorn–Gaussian |
|---|---|---|---|---|---|---|---|---|
| Camelyon17 ($d=2048$) | **0.0214/0.0575** | 6.85/13.8 | 6.62/7.45 | 16.4/17.3 | 19/19 | 8.1/NR (0.0608) | 7.8/NR (0.0991) | 2.23/NR (0.0962) |
| VisDA-2017 ($d=2048$) | **0.0369/0.055** | 7.32/13.2 | 6.22/7.47 | 16.6/17.4 | 19/19 | NR (0.107) | 2.17/NR (0.0652) | 0.877/NR (0.0703) |
| Terra/CCT-20 ($d=2048$) | **0.0188/0.0369** | 5.76/11.5 | 6.82/8.07 | 16.9/17.8 | 19.2/19.2 | 8.28/NR (0.0508) | 1.73/NR (0.0512) | 0.419/NR (0.0345) |

*Table 2.* **CovDrift-MR downstream on Camelyon17 under strict step budgets.** Mean±std over three seeds. $t_5$: adaptation time at $K=5$ seconds (shared preprocessing excluded). Settings: $r=16$, drift rank 16, $s_{\max}=1.70$; shared stabilization is as in Appendix E.1.

| Method | $K=1$ | $K=2$ | $K=5$ | $K=20$ | $t_5$ |
|---|---|---|---|---|---|
| No adapt | 70.22 $\pm2.17$ | 70.22 $\pm2.17$ | 70.22 $\pm2.17$ | 70.22 $\pm2.17$ | — |
| ITSPACE | 73.94 $\pm1.79$ | 74.01 $\pm1.78$ | 74.01 $\pm1.78$ | 74.01 $\pm1.78$ | 0.035 $\pm0.014$ |
| BW-geodesic | 71.92 $\pm2.09$ | 72.86 $\pm1.97$ | 73.54 $\pm1.92$ | 74.01 $\pm1.78$ | 0.015 $\pm0.001$ |
| BW-GD | 73.97 $\pm1.78$ | 73.93 $\pm1.80$ | 73.97 $\pm1.77$ | 74.01 $\pm1.78$ | 0.037 $\pm0.004$ |
| Euclidean | 72.11 $\pm2.10$ | 72.11 $\pm2.10$ | 72.11 $\pm2.10$ | 74.01 $\pm1.78$ | 0.021 $\pm0.006$ |
| Log-Euclidean | 70.56 $\pm1.89$ | 70.91 $\pm1.95$ | 71.92 $\pm2.09$ | 74.01 $\pm1.78$ | 0.013 $\pm0.002$ |
| AIRM | 70.56 $\pm1.89$ | 70.91 $\pm1.95$ | 71.92 $\pm2.09$ | 74.01 $\pm1.78$ | 0.013 $\pm0.002$ |
| Sinkhorn | 72.67 $\pm2.45$ | 72.67 $\pm2.45$ | 72.67 $\pm2.45$ | 72.67 $\pm2.45$ | 0.233 $\pm0.014$ |
| Sinkhorn-Gaussian | 72.88 $\pm2.92$ | 72.88 $\pm2.92$ | 72.88 $\pm2.92$ | 72.88 $\pm2.92$ | 0.164 $\pm0.006$ |

only difference is the covariance-alignment update rule (Appendix E.1).

Table 2 gives the full strict-budget sweep on Camelyon17. ITSPACE recovers a large fraction of the drift-induced drop within one to two updates and reaches the endpoint-reference performance band under small budgets. Table 3 summarizes the same protocol across Camelyon17, VisDA-2017, and Terra at representative budgets. Across datasets, ITSPACE reaches endpoint-level downstream performance within one or two updates and keeps adaptation time small; sample-OT baselines can be competitive in task metric but require larger adaptation time.

**Summary.**

- **Experiment I:** Under a fixed rank budget, ITSPACE reaches both GAP thresholds fastest in $t_{\text{alg}}$ across all evaluated covariance-shift datasets.

- **Experiment II:** Under strict budgets ($K \leq 5$), ITSPACE gives fast covariance adaptation and competitive downstream recovery; the strongest downstream method can depend on the dataset.

- **Stability:** The sufficient-decrease theorem and the inexact-certificate bound align the update with the exact BW evaluator used in Experiment I.

## 6. Conclusion

We presented ITSPACE, a few-step method for rank-constrained covariance alignment under the exact Gaussian optimal-transport objective, equivalently the Bures–Wasserstein (BW) discrepancy between centered Gaussian covariances. The main idea is simple: instead of optimizing over dense covariance matrices, ITSPACE updates a square-root factor $Y$ directly, so every iterate $X_k = Y_k Y_k^\top$ remains positive semidefinite and rank constrained by construction. In this factorization, the BW objective becomes a quadratic term minus a nuclear norm; a polar/Procrustes certificate gives a tight linear minorization of the nuclear norm, leading to a closed-form proximal majorization–minimization update. Thus BW covariance alignment can be used as a lightweight inner-loop primitive when only a small number of valid covariance updates are affordable.

On the theory side, the update is not merely heuristic. We proved a sufficient-decrease guarantee for the true BW ob-

*Table 3.* **CovDrift-MR downstream summary across datasets.** Numbers are mean±std over three seeds. We report representative strict budgets for ITSPACE and the strongest non-ITSPACE method at $K{=}5$ under the same protocol. Full budget sweeps for VisDA-2017 and Terra are in Appendix Table 7.

| Dataset / metric | No adapt | ITSPACE $K{=}2$ | ITSPACE $K{=}5$ | Strongest other at $K{=}5$ | ITSPACE $t_5$ |
|---|---|---|---|---|---|
| Camelyon17 AUROC | $70.22 \pm 2.17$ | $74.01 \pm 1.78$ | $74.01 \pm 1.78$ | BW-GD: $73.97 \pm 1.77$ | $0.035 \pm 0.014$ |
| VisDA-2017 Acc. | $75.68 \pm 2.18$ | $84.26 \pm 1.18$ | $84.26 \pm 1.18$ | BW-GD: $83.82 \pm 1.28$ | $0.029 \pm 0.006$ |
| Terra Acc. | $58.89 \pm 2.53$ | $63.94 \pm 2.43$ | $63.94 \pm 2.43$ | Sinkhorn: $65.00 \pm 1.59$ | $0.024 \pm 0.002$ |

jective under exact polar computation, quantified the effect of finite-precision polar computation through an explicit certificate-gap bound, and characterized the full-rank invertible fixed points. The appendix also records a matched-rank PSD extension and shows why singular targets require care in the choice of polar certificate. These results give a compact optimization picture: ITSPACE preserves the desired matrix structure at every step while making certified progress on the exact Gaussian Wasserstein objective.

Empirically, under a unified exact-BW evaluator and a common rank-budget timing protocol, ITSPACE reaches low BW gaps substantially faster than BW-gradient descent and methods based on other covariance geometries. In CovDrift-MR, these fast covariance updates provide competitive downstream recovery under strict update budgets, with the strongest downstream method depending on the dataset. The scope of the method is deliberately covariance-level: ITSPACE is designed for Gaussian representations, second-order feature summaries, and modules that consume covariance factors, rather than as a replacement for full sample-level OT on arbitrary non-Gaussian distributions. Within that scope, it provides a closed-form, rank-compatible, BW-descending update that can be plugged into domain adaptation, normalization, whitening–recoloring, and other covariance-based learning pipelines. Extending this few-step BW alignment primitive to nonlinear shifts, barycenters, and other structured PSD summaries is a natural direction for future work.

## Acknowledgments

We thank the anonymous reviewers for constructive feedback that helped improve the paper. This work was supported by grant NIH/NHLBI 5R01HL167072.

## Impact Statement

Optimal transport is a core tool for comparing and aligning distributions, with applications spanning domain adaptation, representation learning, generative modeling, and probabilistic inference. In many modern pipelines, distributions are summarized through compact second-order descriptors such as feature covariances; in this regime, Gaussian optimal transport yields an exact, closed-form discrepancy on covariances. This work contributes an efficient iterative primitive for covariance alignment under this exact objective, designed for settings where only a few adaptation steps are affordable and where covariances must remain well-posed throughout the update trajectory (e.g., fast test-time adaptation from unlabeled target batches, or inner-loop modules used inside larger training and deployment systems). By reducing the number of updates required to reach a given alignment quality and by relying on lightweight linear-algebra operations, such methods can lower computational cost, latency, and energy use. Moreover, directly optimizing the exact Bures–Wasserstein loss can yield more faithful alignment toward a target covariance than surrogate geometries or regularized OT solvers, which may translate into improved reliability for downstream components that consume covariance estimates (e.g., whitening/normalization layers or Gaussian embeddings). ITSPACE is a general optimization method and does not encode application-specific intent; like other broadly applicable ML tools, it may be misused by bad actors, and the community is encouraged to deploy distribution-alignment techniques in ways that benefit society.

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

# A. Additional Theory: Proofs and Derivations

## A.1. Validity of the Factor Iterates

*Proof of Proposition 4.1.* For any factor $Y_k \in \mathbb{R}^{d \times r}$, the matrix $X_k = Y_k Y_k^\top$ is symmetric and positive semidefinite. Moreover,

$$\text{rank}(X_k) = \text{rank}(Y_k Y_k^\top) \leq \text{rank}(Y_k) \leq r.$$

If $r = d$ and $Y_k$ is invertible, then $X_k = Y_k Y_k^\top$ is positive definite, so $X_k \in \mathbb{S}_{++}^d$. □

## A.2. Support Function, Polar Certificates, and the Majorizer

We first recall the support-function characterization of the nuclear norm. For any $A \in \mathbb{R}^{d \times r}$,

$$\|A\|_* = \max_{\|Q\|_2 \leq 1} \text{tr}(Q^\top A), \tag{20}$$

where $\| \cdot \|_2$ denotes the operator norm.

**Lemma A.1** (Support-Function Maximizer). *Let $A \in \mathbb{R}^{d \times r}$, and let $A = U\Gamma V^\top$ be a compact singular value decomposition with rank $\rho$. If $\rho > 0$, then $Q^\star = UV^\top$ is feasible for (20) and attains the maximum. If $A = 0$, then every feasible $Q$ with $\|Q\|_2 \leq 1$ is optimal; in particular, $Q = 0$ is a valid choice.*

*Proof.* Assume first that $\rho > 0$. Since $U$ and $V$ have orthonormal columns, $\|UV^\top\|_2 = 1$, so $Q^\star = UV^\top$ is feasible. Moreover,

$$\text{tr}((Q^\star)^\top A) = \text{tr}(VU^\top U\Gamma V^\top) = \text{tr}(\Gamma) = \|A\|_*.$$

Thus $Q^\star$ attains the maximum in (20). If $A = 0$, then $\text{tr}(Q^\top A) = 0 = \|A\|_*$ for every feasible $Q$, so every feasible $Q$ is optimal. □

Throughout the paper, $\text{polar}(A)$ denotes a selected maximizer in (20):

$$\text{polar}(A) \in \arg \max_{\|Q\|_2 \leq 1} \text{tr}(Q^\top A).$$

When $A$ has full column rank, this coincides with the usual thin polar factor $A(A^\top A)^{-1/2}$. When $A$ is rank deficient, the maximizer may be nonunique and need not satisfy $Q^\top Q = I_r$.

**Lemma A.2** (Global Quadratic Majorizer). *Let*

$$F(Y) = \|Y\|_F^2 + \text{tr}(\Sigma_\star) - 2\|SY\|_*, \qquad S = \Sigma_\star^{1/2}.$$

*Fix $Y_k$, set $A_k := SY_k$, choose an exact certificate $Q_k := \text{polar}(A_k)$, and define $B_k := SQ_k$. Then, for every $Y$,*

$$F(Y) \leq \|Y - B_k\|_F^2 + C_k, \qquad C_k := \text{tr}(\Sigma_\star) - \|B_k\|_F^2. \tag{21}$$

*Moreover, the bound is tight at $Y = Y_k$.*

*Proof.* Since $Q_k$ is feasible in (20), for every $Y$,

$$\|SY\|_* \geq \text{tr}(Q_k^\top SY). \tag{22}$$

Since $Q_k$ is an exact maximizer for $A_k = SY_k$, this inequality is tight at $Y = Y_k$:

$$\|SY_k\|_* = \text{tr}(Q_k^\top SY_k). \tag{23}$$

Using (22) in the factor objective gives

$$\begin{aligned} F(Y) &= \|Y\|_F^2 + \text{tr}(\Sigma_\star) - 2\|SY\|_* \\ &\leq \|Y\|_F^2 + \text{tr}(\Sigma_\star) - 2\,\text{tr}(Q_k^\top SY) \\ &= \|Y\|_F^2 + \text{tr}(\Sigma_\star) - 2\,\text{tr}(B_k^\top Y). \end{aligned} \tag{24}$$

Completing the square,

$$\|Y\|_F^2 - 2\,\text{tr}(B_k^\top Y) = \|Y - B_k\|_F^2 - \|B_k\|_F^2,$$

which proves (21). Tightness at $Y = Y_k$ follows from (23). □

### A.3. Sufficient BW Descent under Exact Polar Certificates

*Proof of Theorem 4.2.* Fix $k$. By Lemma A.2, the exact polar certificate induces the tight global upper bound

$$F(Y) \leq \|Y - B_k\|_F^2 + C_k.$$

Adding the proximal term gives

$$U_k(Y) := \|Y - B_k\|_F^2 + C_k + \frac{1}{2\lambda}\|Y - Y_k\|_F^2. \tag{25}$$

Then

$$F(Y) \leq U_k(Y) \quad \text{for all } Y, \qquad U_k(Y_k) = F(Y_k).$$

The minimizer of $U_k$ is the update in Eq. (12). Since

$$\alpha = \frac{2\lambda}{1 + 2\lambda}, \qquad \frac{1}{\alpha} = 1 + \frac{1}{2\lambda},$$

completion of squares gives

$$U_k(Y) = U_k(Y_{k+1}) + \frac{1}{\alpha}\|Y - Y_{k+1}\|_F^2. \tag{26}$$

Evaluating (26) at $Y = Y_k$ gives

$$U_k(Y_k) - U_k(Y_{k+1}) = \frac{1}{\alpha}\|Y_k - Y_{k+1}\|_F^2.$$

Therefore,

$$F(Y_k) - F(Y_{k+1}) \geq U_k(Y_k) - U_k(Y_{k+1}) = \frac{1}{\alpha}\|Y_{k+1} - Y_k\|_F^2.$$

Finally, because

$$Y_{k+1} = \alpha B_k + (1 - \alpha)Y_k,$$

we have

$$Y_{k+1} - Y_k = \alpha(B_k - Y_k),$$

and hence

$$\frac{1}{\alpha}\|Y_{k+1} - Y_k\|_F^2 = \alpha\|B_k - Y_k\|_F^2.$$

This proves Eq. (14). Since $F(Y_k) = W_2^2(Y_k Y_k^\top, \Sigma_\star)$, the BW objective is monotone nonincreasing along the covariance iterates. $\qquad\square$

### A.4. Inexact Polar Certificate Bound

*Proof of Proposition 4.3.* Let $A_k := SY_k$, and suppose $\widehat{Q}_k$ satisfies

$$\|\widehat{Q}_k\|_2 \leq 1, \qquad \mathrm{tr}(\widehat{Q}_k^\top A_k) \geq \|A_k\|_* - \varepsilon_k.$$

Define the inexact surrogate

$$\widehat{U}_k(Y) := \|Y\|_F^2 + \mathrm{tr}(\Sigma_\star) - 2\,\mathrm{tr}(\widehat{Q}_k^\top SY) + \frac{1}{2\lambda}\|Y - Y_k\|_F^2.$$

Feasibility of $\widehat{Q}_k$ gives

$$\|SY\|_* \geq \mathrm{tr}(\widehat{Q}_k^\top SY) \qquad \text{for all } Y,$$

and therefore

$$F(Y) \leq \widehat{U}_k(Y) \qquad \text{for all } Y.$$

At the current iterate,

$$\begin{aligned}
\widehat{U}_k(Y_k) &= \|Y_k\|_F^2 + \mathrm{tr}(\Sigma_\star) - 2\,\mathrm{tr}(\widehat{Q}_k^\top A_k) \\
&\leq \|Y_k\|_F^2 + \mathrm{tr}(\Sigma_\star) - 2\|A_k\|_* + 2\varepsilon_k \\
&= F(Y_k) + 2\varepsilon_k.
\end{aligned} \tag{27}$$

Let $Y_{k+1} = \arg\min_Y \widehat{U}_k(Y)$. The same completion-of-squares identity gives

$$\widehat{U}_k(Y) = \widehat{U}_k(Y_{k+1}) + \frac{1}{\alpha}\|Y - Y_{k+1}\|_F^2.$$

Setting $Y = Y_k$ and using (27), we obtain

$$
\begin{aligned}
F(Y_{k+1}) &\le \widehat{U}_k(Y_{k+1}) \\
&= \widehat{U}_k(Y_k) - \frac{1}{\alpha}\|Y_{k+1} - Y_k\|_F^2 \\
&\le F(Y_k) + 2\varepsilon_k - \frac{1}{\alpha}\|Y_{k+1} - Y_k\|_F^2.
\end{aligned}
\tag{28}
$$

Dropping the nonpositive last term gives

$$F(Y_{k+1}) \le F(Y_k) + 2\varepsilon_k,$$

which proves Eq. (17). When $\varepsilon_k = 0$, (28) recovers the sufficient-descent bound. $\square$

### A.5. Full-Rank Invertible Fixed Points

*Proof of Proposition 4.4.* Assume $r = d$ and $\Sigma_\star \in \mathbb{S}_{++}^d$, and restrict attention to invertible factor iterates. Let $S = \Sigma_\star^{1/2}$.

First, consider any factor of the form

$$Y = SR, \qquad R \in O(d).$$

Then

$$SY = S^2 R = \Sigma_\star R.$$

Since $\Sigma_\star$ is positive definite and $R$ is orthogonal, $\Sigma_\star R$ is invertible, and its polar factor is

$$\operatorname{polar}(\Sigma_\star R) = \Sigma_\star R \big(R^\top \Sigma_\star^2 R\big)^{-1/2} = \Sigma_\star R \big(R^\top \Sigma_\star^{-1} R\big) = R.$$

Substituting this into the update gives

$$Y_{k+1} = \alpha SR + (1 - \alpha)SR = SR,$$

so every $Y = SR$ with $R \in O(d)$ is a fixed point.

Conversely, let $Y$ be an invertible fixed point. Since $S$ is invertible, $SY$ is invertible, and the polar factor

$$R := \operatorname{polar}(SY)$$

is orthogonal. The fixed-point equation gives

$$Y = \alpha S \operatorname{polar}(SY) + (1 - \alpha)Y.$$

Since $\alpha \in (0, 1)$, this implies

$$Y = S \operatorname{polar}(SY) = SR.$$

Therefore every invertible factor fixed point is of the form $Y = SR$ with $R \in O(d)$. Its covariance is

$$YY^\top = SRR^\top S = S^2 = \Sigma_\star.$$

Thus the corresponding covariance fixed point in the full-rank invertible stratum is uniquely $X = \Sigma_\star$. $\square$

### A.6. Equivariance under Consistent Certificate Selection

Because the support-function maximizer can be nonunique for rank-deficient matrices, equivariance statements require a consistent selection of polar certificates. This condition is automatic when the relevant polar certificate is unique, such as in the full-rank square case.

**Proposition A.3** (Equivariance under Consistent Certificate Selection)**.** *Let $U$ be an orthogonal matrix and let $c > 0$.*

1. *Suppose that, under the transformed initialization*

$$\Sigma_\star^{(U)} = U\Sigma_\star U^\top, \qquad Y_0^{(U)} = UY_0,$$

   *the polar certificates are selected consistently as*

$$Q_k^{(U)} = UQ_k \qquad \text{whenever } Y_k^{(U)} = UY_k.$$

   *Then the covariance trajectory satisfies*

$$X_k^{(U)} = UX_k U^\top \qquad \text{for all } k.$$

2. *Suppose that, under the rescaled initialization*

$$\Sigma_\star^{(c)} = c\Sigma_\star, \qquad Y_0^{(c)} = \sqrt{c}\, Y_0,$$

   *the polar certificates are selected consistently as*

$$Q_k^{(c)} = Q_k \qquad \text{whenever } Y_k^{(c)} = \sqrt{c}\, Y_k.$$

   *Then the covariance trajectory satisfies*

$$X_k^{(c)} = cX_k \qquad \text{for all } k.$$

*Proof.* For the orthogonal change of basis, the target square root transforms as

$$S^{(U)} = (U\Sigma_\star U^\top)^{1/2} = USU^\top.$$

Assume inductively that $Y_k^{(U)} = UY_k$. Then

$$S^{(U)}Y_k^{(U)} = USU^\top UY_k = U(SY_k).$$

Under the stated consistent certificate choice, $Q_k^{(U)} = UQ_k$. Therefore,

$$Y_{k+1}^{(U)} = \alpha S^{(U)}Q_k^{(U)} + (1-\alpha)Y_k^{(U)} = \alpha USQ_k + (1-\alpha)UY_k = UY_{k+1}.$$

Thus

$$X_k^{(U)} = Y_k^{(U)}(Y_k^{(U)})^\top = UX_k U^\top$$

for all $k$.

For global rescaling, the target square root is

$$S^{(c)} = (c\Sigma_\star)^{1/2} = \sqrt{c}\, S.$$

Assume inductively that $Y_k^{(c)} = \sqrt{c}\, Y_k$. Then

$$S^{(c)}Y_k^{(c)} = cSY_k.$$

Since $c > 0$, the support-function maximizers for $SY_k$ and $cSY_k$ coincide as sets. Under the stated consistent certificate choice, $Q_k^{(c)} = Q_k$. Hence

$$Y_{k+1}^{(c)} = \alpha S^{(c)}Q_k^{(c)} + (1-\alpha)Y_k^{(c)} = \sqrt{c}\left(\alpha SQ_k + (1-\alpha)Y_k\right) = \sqrt{c}\, Y_{k+1}.$$

Thus

$$X_k^{(c)} = Y_k^{(c)}(Y_k^{(c)})^\top = cX_k$$

for all $k$. $\qquad\square$

## A.7. Matched-Rank PSD Targets and Certificate Choice

The main text assumes $\Sigma_\star \in \mathbb{S}_{++}^d$. We record here a matched-rank PSD extension. The result shows that a linear contraction holds when the target has rank $r$ and the polar certificate is chosen in the target subspace. The example after the proposition shows why this certificate condition is necessary.

**Proposition A.4** (Matched-Rank PSD Contraction). *Assume* $\mathrm{rank}(\Sigma_\star) = r$ *and write*

$$\Sigma_\star = U_r \Lambda U_r^\top, \qquad T := U_r \Lambda^{1/2} \in \mathbb{R}^{d \times r},$$

*where* $U_r^\top U_r = I_r$ *and* $\Lambda \in \mathbb{R}^{r \times r}$ *is positive definite. Define*

$$\widetilde{F}(Y) := \|Y\|_F^2 + \|T\|_F^2 - 2\|T^\top Y\|_* = \min_{R \in O(r)} \|Y - TR\|_F^2. \tag{29}$$

*Let*

$$R_k \in \arg \max_{R \in O(r)} \mathrm{tr}(R^\top T^\top Y_k),$$

*and choose the exact polar certificate*

$$Q_k = U_r R_k.$$

*Then the* ITSPACE *update satisfies*

$$\widetilde{F}(Y_{k+1}) \leq (1 - \alpha)^2 \widetilde{F}(Y_k). \tag{30}$$

*Consequently,*

$$\widetilde{F}(Y_k) \leq (1 - \alpha)^{2k} \widetilde{F}(Y_0).$$

*Proof.* Let

$$S = \Sigma_\star^{1/2} = U_r \Lambda^{1/2} U_r^\top = T U_r^\top.$$

For any $Y \in \mathbb{R}^{d \times r}$,

$$\|SY\|_* = \|U_r \Lambda^{1/2} U_r^\top Y\|_* = \|\Lambda^{1/2} U_r^\top Y\|_* = \|T^\top Y\|_*,$$

because left multiplication by $U_r$ preserves the nonzero singular values. Thus $\widetilde{F}$ is the rank-matched PSD analogue of the factor objective.

The equality in (29) follows from the orthogonal Procrustes identity:

$$\min_{R \in O(r)} \|Y - TR\|_F^2 = \|Y\|_F^2 + \|T\|_F^2 - 2 \max_{R \in O(r)} \mathrm{tr}(R^\top T^\top Y) = \widetilde{F}(Y).$$

By definition of $R_k$, $Q_k = U_r R_k$ is feasible with $\|Q_k\|_2 = 1$, and

$$\mathrm{tr}(Q_k^\top S Y_k) = \mathrm{tr}(R_k^\top U_r^\top U_r \Lambda^{1/2} U_r^\top Y_k) = \mathrm{tr}(R_k^\top T^\top Y_k) = \|T^\top Y_k\|_* = \|SY_k\|_*.$$

Thus $Q_k$ is an exact support-function certificate. Moreover,

$$B_k = SQ_k = U_r \Lambda^{1/2} U_r^\top U_r R_k = TR_k.$$

The update therefore becomes

$$Y_{k+1} = \alpha TR_k + (1 - \alpha) Y_k.$$

Using the same $R_k$ as a feasible comparison in the Procrustes form of $\widetilde{F}(Y_{k+1})$, we obtain

$$\begin{aligned}
\widetilde{F}(Y_{k+1}) &= \min_{R \in O(r)} \|Y_{k+1} - TR\|_F^2 \\
&\leq \|Y_{k+1} - TR_k\|_F^2 \\
&= \|(1 - \alpha)(Y_k - TR_k)\|_F^2 \\
&= (1 - \alpha)^2 \|Y_k - TR_k\|_F^2 \\
&= (1 - \alpha)^2 \widetilde{F}(Y_k). \tag{31}
\end{aligned}$$

Iterating this inequality proves the final claim. $\square$

*Remark* A.5 (Why the Certificate Condition Is Necessary). Consider

$$\Sigma_\star = \mathrm{diag}(1,1,0), \qquad r = 2, \qquad Y_0 = [e_1,\ 0],$$

where $e_i$ denotes the $i$th coordinate vector in $\mathbb{R}^3$. Then $S = \Sigma_\star^{1/2} = \Sigma_\star$ and

$$SY_0 = [e_1,\ 0].$$

The matrix

$$Q_0 = [e_1,\ e_3]$$

is feasible with $\|Q_0\|_2 = 1$, and it is an exact support-function maximizer because

$$\mathrm{tr}(Q_0^\top SY_0) = 1 = \|SY_0\|_*.$$

However,

$$B_0 = SQ_0 = [e_1,\ 0] = Y_0,$$

so the update gives

$$Y_1 = \alpha B_0 + (1-\alpha)Y_0 = Y_0.$$

Thus the iteration stalls even though

$$Y_0 Y_0^\top = \mathrm{diag}(1,0,0) \neq \mathrm{diag}(1,1,0) = \Sigma_\star.$$

This example shows that, in singular cases, exactness of the polar certificate alone is not enough to guarantee the matched-rank contraction; the certificate must also be chosen consistently with the target subspace.

### A.8. Derivation of the Factor Form

*Derivation of Eq.* (5). Let $X = YY^\top$ and $S = \Sigma_\star^{1/2}$. From the BW closed form in Eq. (2),

$$W_2^2(YY^\top, \Sigma_\star) = \mathrm{tr}(YY^\top) + \mathrm{tr}(\Sigma_\star) - 2\,\mathrm{tr}\Big((SYY^\top S)^{1/2}\Big).$$

We have $\mathrm{tr}(YY^\top) = \|Y\|_F^2$, and $SYY^\top S = (SY)(SY)^\top$. Let $A := SY$. Then

$$\mathrm{tr}\Big((SYY^\top S)^{1/2}\Big) = \mathrm{tr}\Big((AA^\top)^{1/2}\Big) = \|A\|_* = \|SY\|_*.$$

Substituting these identities yields

$$W_2^2(YY^\top, \Sigma_\star) = \|Y\|_F^2 + \mathrm{tr}(\Sigma_\star) - 2\|SY\|_*,$$

which is Eq. (5). $\qquad\square$

### A.9. Commuting-Case Justification for the Rank-$r$ Floor

**Lemma A.6** (Commuting Case). *Let $\Pi_r(\Sigma_\star)$ be the rank-r truncation of $\Sigma_\star$ using the top r eigencomponents. Then*

$$W_2^2(\Pi_r(\Sigma_\star), \Sigma_\star) = \sum_{i=r+1}^{d} \lambda_i(\Sigma_\star),$$

*where $\lambda_1(\Sigma_\star) \geq \cdots \geq \lambda_d(\Sigma_\star) \geq 0$.*

*Proof.* Since $\Pi_r(\Sigma_\star)$ and $\Sigma_\star$ share eigenvectors, there exists an orthogonal basis in which both are diagonal:

$$\Sigma_\star = \mathrm{diag}(\lambda_1, \ldots, \lambda_d), \qquad \Pi_r(\Sigma_\star) = \mathrm{diag}(\lambda_1, \ldots, \lambda_r, 0, \ldots, 0).$$

For diagonal matrices, the BW closed form reduces to

$$W_2^2(X, \Sigma_\star) = \sum_{i=1}^d \lambda_i(X) + \sum_{i=1}^d \lambda_i(\Sigma_\star) - 2\sum_{i=1}^d \sqrt{\lambda_i(X)\lambda_i(\Sigma_\star)}.$$

Substituting $\lambda_i(X) = \lambda_i(\Sigma_\star)$ for $i \leq r$ and $\lambda_i(X) = 0$ for $i > r$ gives

$$W_2^2(\Pi_r(\Sigma_\star), \Sigma_\star) = \sum_{i=r+1}^d \lambda_i(\Sigma_\star).$$

$\square$

## B. Implementation Details

This section documents practical choices for ITSPACE and for enforcing rank budgets.

### B.1. Initialization and Factor Handling

We maintain $Y_k \in \mathbb{R}^{d \times r}$ and materialize $X_k = Y_k Y_k^\top$ only when needed. Common initializations include:

- **From a PSD covariance $X_0$:** compute a square-root factor $Y_0$ via an eigendecomposition (or Cholesky when SPD).

- **Rank-$r$ from an SPD covariance:** if $X_0 \approx U_r \Lambda_r U_r^\top$ (top-$r$ eigenpairs), set $Y_0 = U_r \Lambda_r^{1/2}$.

- **From centered features:** if $X_0 = \frac{1}{n-1} Z^\top Z$ with centered $Z \in \mathbb{R}^{n \times d}$, then $Y_0 = \frac{1}{\sqrt{n-1}} Z^\top$ is a valid factor without forming $X_0$ explicitly.

### B.2. Polar Computation and Numerical Stabilization

At iteration $k$, ITSPACE forms $A_k := SY_k \in \mathbb{R}^{d \times r}$ and a certificate $Q_k := \mathrm{polar}(A_k) \in \arg\max_{\|Q\|_2 \leq 1} \mathrm{tr}(Q^\top A_k)$.

**SVD route.** Compute a compact SVD $A_k = U\Sigma V^\top$ and set $Q_k := UV^\top$.

**Gram route (when $r \ll d$).** Form $G_k := A_k^\top A_k \in \mathbb{S}_+^r$ and set $Q_k := A_k G_k^{-1/2}$. If $G_k$ is ill-conditioned, use $G_{k,\delta} := G_k + \delta I$ and $Q_k := A_k G_{k,\delta}^{-1/2}$ with $\delta > 0$ (e.g., proportional to $\mathrm{tr}(G_k)/r$).

**Feasibility normalization.** For the inexact-certificate bound (Proposition 4.3), it suffices that $\|\widehat{Q}_k\|_2 \leq 1$. If a numerical routine returns $\|\widehat{Q}_k\|_2 > 1$, enforce feasibility by

$$\widehat{Q}_k \leftarrow \widehat{Q}_k \Big/ \max\{1, \|\widehat{Q}_k\|_2\}.$$

**SPD vs. PSD in practice.** The theory assumes $\Sigma_\star \in \mathbb{S}_{++}^d$. In experiments, when a baseline requires SPD and an estimated covariance is nearly singular, we apply a small diagonal floor (Appendix D.1).

### B.3. Certificate Gap and Diagnostics

Given $A_k = SY_k$ and a feasible certificate $\widehat{Q}_k$,

$$\varepsilon_k := \|A_k\|_* - \mathrm{tr}(\widehat{Q}_k^\top A_k) \geq 0.$$

We also monitor $\|\widehat{Q}_k^\top \widehat{Q}_k - I\|_F$ (when applicable) as a numerical diagnostic.

### B.4. Complexity Breakdown

**One-time preprocessing.** Computing $S = \Sigma_\star^{1/2}$ is $O(d^3)$ time and $O(d^2)$ memory and is performed once per instance.

*Table 4.* **Covariance-alignment baselines: objective targeted, guarantees, and complexity.** We indicate whether a method targets the exact Gaussian OT objective (2), whether it provides a monotonicity guarantee for the evaluated BW energy $W_2^2(\cdot, \Sigma_\star)$, and its dense per-step cost. In rank-budget runs, full-rank methods are followed by a rank-$r$ truncation, adding a projection term $\text{proj}(d, r)$ that is counted in $t_{\text{alg}}$.

| Method | Objective / update rule | Targets Gaussian $W_2^2$? | Monotone $W_2^2(\cdot, \Sigma_\star)$? | Cost / step |
|---|---|---|---|---|
| ITSPACE | Proximal MM updates for BW in a square-root factor-ization; polar/Procrustes certificate + damping | Yes | Yes[†] | $O(d^2 r + dr^2 + r^3)$ |
| BW geodesic | Closed-form BW geodesic interpolation toward $\Sigma_\star$ (displacement interpolation), then rank-$r$ truncation in rank-budget runs | Yes | Yes (in geodesic parameter) | $O(d^3) + \text{proj}(d, r)$ |
| BW-GD (direct BW optimizer) | BW-targeting gradient/Riemannian updates with optional backtracking, then rank-$r$ truncation in rank-budget runs | Yes | Not in general[⋆] | $O(d^3) + \text{proj}(d, r)$ |
| Euclidean / Frobenius | Ambient-space interpolation/update, then rank-$r$ truncation in rank-budget runs | No | No | $O(d^2) + \text{proj}(d, r)$ |
| Log–Euclidean | Log-domain geodesic / Log–Euclidean objective (full-matrix functions), then truncation | No | No | $O(d^3) + \text{proj}(d, r)$ |
| AIRM | Affine-invariant metric geodesic / AIRM objective (full-matrix functions), then truncation | No | No | $O(d^3) + \text{proj}(d, r)$ |
| CORAL | Whitening/re-coloring (second-moment match); used as a one-shot endpoint reference in downstream | No[‡] | N/A | $O(d^3) + \text{proj}(d, r)$ |
| Sinkhorn (entropic OT) | Regularized OT between samples (biased objective) | No (regularized) | No | $O(Tnm)$ |
| Sinkhorn–Gaussian | Regularized OT on Gaussian samples/embeddings (biased objective) | No (regularized) | No | $O(Tnm)$ |

[†] In exact arithmetic with an exact polar certificate. [⋆] Gradient methods can be made descending with sufficiently small step sizes and/or line search, but do not provide an inherent monotonicity certificate for BW without additional control. [‡] CORAL matches covariances in closed form in the unconstrained full-rank setting, but is not derived as a BW-descent method and does not address the few-step rank-budget regime. Here $\text{proj}(d, r)$ denotes the cost of rank-$r$ PSD truncation (implementation-dependent; included in $t_{\text{alg}}$).

*Table 5.* **Datasets and shifts.** All datasets use covariance dimension $d=2048$ and, unless stated otherwise, rank budget $r=16$, step budget $K=20$, and three seeds $\{0, 1, 2\}$.

| Dataset | Shift (source → target) |
|---|---|
| Camelyon17 (WILDS) (Koh et al., 2021) | hospitals (train → test) |
| VisDA-2017 (Peng et al., 2017) | synthetic → real (train → validation) |
| Terra Incognita / CCT-20 (Beery et al., 2018) | location shift (CIS/TRANS) |

**Per-iteration update.** Each iteration forms $A_k = SY_k$ and $SQ_k$ in $O(d^2 r)$ time. The polar step costs: (i) $O(dr^2)$ via compact SVD when $r \ll d$, or (ii) $O(dr^2 + r^3)$ via the Gram route. Overall per-iteration update complexity is $O(d^2 r + dr^2 + r^3)$.

## C. Baselines: Taxonomy, Guarantees, and Cost

Table 4 summarizes the baseline families used in this paper. In rank-budget experiments, full-rank baselines are followed by a rank-$r$ truncation $\Pi_r(\cdot)$; the corresponding projection time is included in $t_{\text{alg}}$.

## D. Experimental Protocol and Additional Results

### D.1. Datasets and Covariance Construction

**Covariance Construction.** Given features $Z \in \mathbb{R}^{n \times d}$ whose rows are samples, we form the centered covariance

$$\widehat{\Sigma} = \frac{1}{n-1}(Z - \bar{Z})^\top (Z - \bar{Z}),$$

and symmetrize via

$$\widehat{\Sigma} \leftarrow \frac{1}{2}(\widehat{\Sigma} + \widehat{\Sigma}^\top).$$

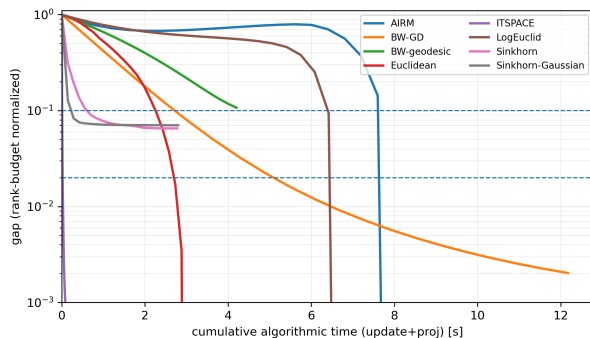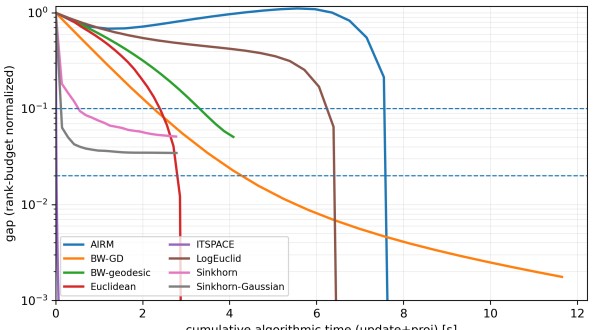

*Figure 3.* **Larger-format rank-budget GAP contraction plots.** Left: VisDA-2017 synthetic $\rightarrow$ real, $d{=}2048$, $r{=}16$, $K{=}20$. Right: Terra/CCT-20 CIS/TRANS, $d{=}2048$, $r{=}16$, $K{=}20$. Curves plot $\mathrm{gap}_r$ versus $t_{\mathrm{alg}} = t_{\mathrm{update}} + t_{\mathrm{proj}}$, excluding shared BW-evaluation overhead; dashed lines mark $\tau \in \{0.1, 0.02\}$.

**SPD Handling When Required.** When a baseline requires SPD input and $\lambda_{\min}(\widehat{\Sigma}) \leq 0$, we use

$$\widehat{\Sigma} \leftarrow \widehat{\Sigma} + \delta I, \qquad \delta = -\lambda_{\min}(\widehat{\Sigma}) + 10^{-8}.$$

The same stabilization rule is applied across methods whenever it is needed.

**Rank-$r$ Truncation.** Rank-budgeted evaluation uses $\Pi_r(\cdot)$, the top-$r$ PSD truncation. When a baseline produces full-rank iterates, we apply $\Pi_r$ after each update; this projection time is included in $t_{\mathrm{alg}}$.

### D.2. Timing Protocol

We decompose wall-clock time into three components: update time, projection time, and BW-evaluation time. Projection time refers to explicit rank-$r$ truncation when used. BW-evaluation time refers to exact BW computation used only for logging or GAP calculation.

**What Is Included in $t_{\mathrm{alg}}$.** The main-text time axis is cumulative algorithmic time

$$t_{\mathrm{alg}} = t_{\mathrm{update}} + t_{\mathrm{proj}}.$$

Method-internal checks, such as BW-GD backtracking or line-search evaluations, are counted in $t_{\mathrm{update}}$. For methods that produce full-rank iterates, projection time is included because the evaluated covariance state is required to satisfy the shared rank budget.

**What Is Excluded from $t_{\mathrm{alg}}$.** Shared BW-evaluation time for logging is excluded. One-time target preprocessing that is constant across the inner loop, such as computing $\Sigma_\star^{1/2}$, is also excluded unless stated otherwise.

**Measurement Details.** For the covariance-alignment timing in Experiment I, runtimes are synchronized single-GPU wall-clock times measured on an NVIDIA RTX A6000 with an AMD EPYC 7763 CPU and approximately 503 GiB RAM. Runs are single-device and sequential across methods and seeds, not distributed or multi-GPU. We use `time.perf_counter`; for GPU runs, timing blocks are bracketed by `torch.cuda.synchronize`. Each configuration is run for seeds $\{0, 1, 2\}$, and we report medians for timing. CovDrift-MR adaptation times are reported under the corresponding canonical downstream protocol and are used for within-protocol method comparisons.

### D.3. Additional Rank-Budget Contraction Plots

Figure 3 provides larger-format versions of the VisDA-2017 and Terra/CCT-20 GAP contraction plots shown in Figure 2. The protocol is the same as in the main text: exact BW evaluator, $r{=}16$, $K{=}20$, and three seeds.

*Table 6.* **Speedup over BW-GD across datasets.** Ratios are computed from median $t_{\text{alg}}$ values in the time-to-gap protocol. All settings use $r=16$, $K=20$, and three seeds.

| Dataset | speedup at gap $\leq 0.1$ | speedup at gap $\leq 0.02$ |
|---|---|---|
| Camelyon17 ($d=2048$) | 319.9× | 239.7× |
| VisDA-2017 ($d=2048$) | 198.2× | 239.8× |
| Terra/CCT-20 ($d=2048$) | 306.9× | 310.8× |

*Table 7.* **CovDrift-MR downstream on VisDA-2017 and Terra** (accuracy, %). Mean±std over three seeds. $t_5$ is method-specific adaptation time at $K=5$ seconds, excluding shared preprocessing. The protocol and hyperparameters match Table 2 in the main text.

| Method | VisDA-2017 (Accuracy, %) | | | | | Terra (Accuracy, %) | | | | |
|---|---|---|---|---|---|---|---|---|---|---|
| | $K=1$ | $K=2$ | $K=5$ | $K=20$ | $t_5$ (s) | $K=1$ | $K=2$ | $K=5$ | $K=20$ | $t_5$ (s) |
| No adapt | $75.68 \pm 2.18$ | $75.68 \pm 2.18$ | $75.68 \pm 2.18$ | $75.68 \pm 2.18$ | $< 0.001$ | $58.89 \pm 2.53$ | $58.89 \pm 2.53$ | $58.89 \pm 2.53$ | $58.89 \pm 2.53$ | — |
| ITSPACE | $83.93 \pm 1.18$ | $84.26 \pm 1.18$ | $84.26 \pm 1.18$ | $84.26 \pm 1.18$ | $0.029 \pm 0.006$ | $63.83 \pm 2.36$ | $63.94 \pm 2.43$ | $63.94 \pm 2.43$ | $63.94 \pm 2.43$ | $0.024 \pm 0.002$ |
| BW-geodesic | $77.04 \pm 0.76$ | $78.14 \pm 0.85$ | $80.95 \pm 0.55$ | $84.26 \pm 1.18$ | $0.013 \pm 0.000$ | $59.78 \pm 2.78$ | $60.44 \pm 2.94$ | $62.00 \pm 2.62$ | $63.94 \pm 2.43$ | $0.012 \pm 0.001$ |
| BW-GD | $81.00 \pm 1.10$ | $82.82 \pm 1.09$ | $83.82 \pm 1.28$ | $84.26 \pm 1.18$ | $0.039 \pm 0.011$ | $62.22 \pm 2.53$ | $63.56 \pm 2.68$ | $63.89 \pm 2.75$ | $63.94 \pm 2.43$ | $0.029 \pm 0.004$ |
| Euclidean | $78.18 \pm 1.16$ | $79.57 \pm 1.34$ | $81.39 \pm 1.32$ | $84.26 \pm 1.18$ | $0.013 \pm 0.000$ | $60.72 \pm 3.08$ | $61.56 \pm 2.36$ | $62.72 \pm 2.36$ | $63.94 \pm 2.43$ | $0.012 \pm 0.000$ |
| Log-Euclidean | $76.44 \pm 0.93$ | $77.09 \pm 1.18$ | $79.37 \pm 1.18$ | $84.26 \pm 1.18$ | $0.018 \pm 0.008$ | $59.33 \pm 2.57$ | $59.89 \pm 2.76$ | $60.78 \pm 2.77$ | $63.94 \pm 2.43$ | $0.016 \pm 0.007$ |
| AIRM | $76.44 \pm 0.93$ | $77.09 \pm 1.18$ | $79.37 \pm 1.18$ | $84.26 \pm 1.18$ | $0.011 \pm 0.001$ | $59.33 \pm 2.57$ | $59.89 \pm 2.76$ | $60.78 \pm 2.77$ | $63.94 \pm 2.43$ | $0.011 \pm 0.001$ |
| Sinkhorn | $83.50 \pm 1.11$ | $83.50 \pm 1.11$ | $83.50 \pm 1.11$ | $83.50 \pm 1.11$ | $0.219 \pm 0.025$ | $65.00 \pm 1.59$ | $65.00 \pm 1.59$ | $65.00 \pm 1.59$ | $65.00 \pm 1.59$ | $0.179 \pm 0.008$ |
| Sinkhorn-Gaussian | $83.71 \pm 0.89$ | $83.71 \pm 0.89$ | $83.71 \pm 0.89$ | $83.71 \pm 0.89$ | $0.164 \pm 0.010$ | $63.56 \pm 2.61$ | $63.56 \pm 2.61$ | $63.56 \pm 2.61$ | $63.56 \pm 2.61$ | $0.164 \pm 0.004$ |

# E. Additional Downstream Details and Results

## E.1. Downstream: CovDrift-MR Protocol

**Purpose.** CovDrift-MR isolates second-order covariance test-time adaptation under strict step and compute budgets. The predictor is a fixed linear head; adaptation uses only unlabeled target samples for moment estimation; methods differ only in the covariance-alignment update.

**Fixed Head and Shared Preprocessing.** For each seed $s \in \{0, 1, 2\}$, we fit a source-only `StandardScaler` on $Z_s^{\text{train}}$ and train a linear classifier on standardized source features. The head is trained once per seed and frozen for all methods and budgets.

**Matched-Rank Drift.** We inject a stationary rank-$r$ deformation into target features, with $r = 16$ and severity $s_{\max} = 1.70$. The drift is applied to $Z_t^{\text{unlab}}$ for moment estimation and to $Z_t^{\text{test}}$ for evaluation; source features are not drifted.

**Moment Estimation and Stabilization.** Source and target moments $(\mu_s, \Sigma_s)$ and $(\mu_t, \Sigma_t)$ are estimated empirically from source training features and drifted unlabeled target features. All methods share identical stabilization: shrinkage $\gamma = 0.05$ and eigenvalue floor

$$\lambda_{\min} = 10^{-4} \cdot \text{tr}(\widehat{\Sigma})/d.$$

**Budgeted Adaptation and Evaluation.** Each method aligns $\Sigma_s$ toward $\Sigma_t$ under rank budget $r = 16$ and budgets $K \in \{1, 2, 5, 20\}$. From the aligned covariance estimate, we apply a shared whitening–recoloring map to drifted target-test features and evaluate with the frozen head.

**Metrics and Timing.** We report AUROC (%) on Camelyon17 and accuracy (%) on VisDA-2017 and Terra. We report method-specific adaptation time $t_5$ at $K = 5$ seconds, excluding shared preprocessing.

**Terra Closed-Set Evaluation.** For Terra, we evaluate only target-test samples whose labels appear in source training. These labels are used only for evaluation and are handled identically across methods.

## E.2. CovDrift-MR Results on VisDA-2017 and Terra

Table 7 reports the full budget sweeps underlying the cross-dataset downstream summary in Table 3.

