# OpenReview forum: "ITSPACE: Monotone Gaussian Optimal Transport Updates"
_ICML.cc/2026/Conference — ICML 2026 regular_

### Official Review · Reviewer_tLWT · 2026-03-05

**Soundness:** 4
**Presentation:** 4
**Significance:** 3
**Originality:** 3
**Overall Recommendation:** 5
**Confidence:** 4

**Summary:**

This paper proposes an efficient algorithm updating a covariance matrix to make it closer to a target covariance matrix in the Bures--Wasserstein distance under rank constraints.
The algorithm is based on a tight majorization of the objective, coupled with a proximal step.

**Compliance With Llm Reviewing Policy:**

Affirmed.

**Final Justification:**

The discussion round was productive and the authors made a solid impression.
I improve my score.

**Key Questions For Authors:**

1) I get the point of the constraints made at the beginning of the third paragraph of the Introduction. The way it is written appears to fix the issue at a conceptual level. Is there a practical situation with high-dimensional covariance matrices where you encountered the problem? Could you have a use case there that would convince the reader that there is a problem to be tackled?

2) Imagine that the target covariance matrix $\Sigma_*$ has rank $r$. Could you prove that the algorithm converges? If so at what rate?

3) Even though you clearly specify that you focus on covariance matrices, aligning datasets is the ultimate objective. It might be good to discuss the fact that the Wasserstein distance between distributions $P,Q$ can be lower bounded by Bures--Wasserstein distance between Gaussians with the mean vectors and covariance matrices of $P$ and $Q$. In the case of discrete measures, seeing how the improvements at each step translate in terms of the point clouds might be an interesting numerical example to add.

4) In recent years, Bures--Wasserstein distances have been used for graph alignment purposes by seeing graph Laplacians as covariance matrices of Gaussians, and a strong line of work delved into Bures--Wasserstein Barycenters. Is there any way in which you could expand on what you do to serve broader purposes? Can your algorithm be used in a larger pipeline to compute a rank-constrained Bures--Wasserstein barycenter? Would there be applications where this would help summarize datasets?

**Limitations:**

yes

**Strengths And Weaknesses:**

Soundness: The paper is sound. The claims are well supported and easy to follow. Experiments support the claim. This being said, the experiments are relying on only three seeds, which is extremely small. The drift in the experiment is set to 16 which equates the rank of the parametrization of the updated covariance matrix. This seems far from a real-world use case.

Presentation: The paper is easy to read, concise and well-written. It anchors itself well in the literature, even though a more careful review could help sharpen the impact. The tables could be made easier to read by showing what method is best for each setting.

Significance: I believe that the paper has a lot of potential. The scope is quite specialized, but there is a place for such contributions. Further, I have the impression that the authors could be more ambitious than they currently are to improve further the significance of their paper, see my questions/suggestions below.

Original: To the best of my knowledge, the work is original.

---

> ### Author Rebuttal · Authors · 2026-03-31
>
> We sincerely appreciate the thoughtful, encouraging, and constructive review. Below we address the comments and questions.
>
> **Q1:** Yes. A concrete use case is fast adaptation of high-dimensional feature covariances inside larger pipelines. This already appears in practice in correlation alignment for domain adaptation (CORAL / Deep CORAL) [1], in whitening--recoloring style-transfer pipelines (where features are first whitened using the current covariance and then recolored using a target covariance) [2], and in Gaussian transport formulations for generative modeling such as Gaussian Schr\"odinger bridges [3] and Wasserstein flow matching [4].
>
> In these settings, the covariance is not an offline object that can simply be replaced once; it is a high-dimensional state estimated from current features, updated under tight compute/latency constraints, and then consumed immediately downstream. This is exactly the regime instantiated in our paper: 2048-dimensional feature covariances, unlabeled target-batch adaptation, and only $K \in $ {$1,2,5,20$} updates before the aligned covariance is used in the shared whitening--recoloring pipeline of CovDrift-MR.
>
> The practical issue is therefore not whether one can replace a covariance in principle, but whether one can make meaningful progress in a few cheap steps while keeping every intermediate iterate a valid covariance.
>
> [1] Sun et al., CORrelation Alignment for Unsupervised Domain Adaptation, 2017.
>
> [2] Li et al., Universal Style Transfer via Feature Transforms, NeurIPS 2017.
>
> [3] Bunne et al., The Schr\"odinger Bridge between Gaussian Measures has a Closed Form, AISTATS 2023.
>
> [4] Haviv et al., Wasserstein Flow Matching: Generative Modeling Over Families of Distributions, ICML 2025.
>
> **Q2:** For the special case $\operatorname{rank}(\Sigma^\star)=r<d$, the current paper does not claim a general convergence or rate theorem, hence we do not want to overstate this.
>
> What the paper does prove is:
>
>  (i) monotone non-increase of the exact BW objective under exact polar computation
>
>  (ii) an explicit $2\varepsilon_k$ deviation bound under inexact polar certificates
>
>  (iii) a fixed-point characterization in the full-rank SPD case $r=d$.
>
> In that full-rank setting, Sec. 3.2 / Prop. 4.4 show that in the limit $\alpha\to 1$, ITSPACE recovers a factor of $\Sigma^\star$ in one step, and that $X=\Sigma^\star$ is the unique fixed point in covariance space. Extending this analysis to rank-deficient PSD targets and rates is a natural next step, but it is beyond the current claims of the paper.
>
> **Q3:** Thank you for this insightful suggestion. We agree this broader connection is worth making explicit. In particular, discussing the Gaussian-moment lower-bound perspective would better situate the covariance-level BW objective within the broader dataset-alignment picture, and a simple discrete point-cloud illustration would also help make this intuition concrete. We will gladly add this broader discussion in our paper.
>
> **Q4:** Thank you for your comment. We agree this is a valuable broader perspective, and we will add this in the introduction of the revised version to further motivate this BW update primitive within a broader context. Within the current scope, we think the right framing is that ITSPACE is a building block for broader BW-based procedures, including plausible use inside graph-alignment or rank-constrained barycenter pipelines.
>
> **(Broader purposes)** The paper solves the single-target BW covariance-alignment subproblem and gives a closed-form, PSD-preserving, rank-compatible update with monotone descent under exact polar computation. That makes it a plausible inner-loop primitive wherever one repeatedly needs BW covariance updates under a rank budget.
>
> **(Rank-constrained BW barycenter)** Yes, potentially as part of a larger pipeline. A natural route would be to formulate the rank-constrained BW barycenter objective and then use ITSPACE-style single-target rank-compatible updates inside an alternating or iterative optimization scheme, where each step updates the current estimate toward one target covariance at a time while preserving PSD structure and the rank budget. We do not currently claim a full barycenter algorithm or theory in the paper, but we agree this is a natural and promising extension.
>
> **(Dataset summarization)** Yes, this is also a plausible application direction. When datasets are summarized through covariance / Gaussian summaries, a rank-constrained BW primitive could help maintain or update those summaries under compute constraints. We will gladly add this broader discussion in the paper.

---

> > ### Author Rebuttal · Reviewer_tLWT · 2026-04-04
> >
> > Thank you for your answers.
> >
> > I'd like to go back to question 2. Imagine that by chance or the deeds of an oracle, you pick the rank constraint $r$ perfectly and $\Sigma_*$ has that rank. It is quite natural to expect that the algorithm should converge as the rank constraint can be satisfied.
> > I checked the sketch of proof of theorem 4.3. Is it unlikely that one could quantify the gap between $U_k(Y_{k+1})$ and  $U_k(Y_k)$? That could be the start to understanding the convergence rates and, given the form of the functions does not seem far-fetched. Don't hesitate to prove me wrong and guide me through the difficulties!

---

> > > ### Author Response · Authors · 2026-04-06
> > >
> > > Thank you for your response. Under the paper’s stated assumptions, the target covariance satisfies $\Sigma_\star\in\mathbb{S}^d_{++}$, so it is SPD and therefore full rank. Thus your condition $rank(\Sigma_\star)=r$ lies within the formal regime only when $r=d$. For the corresponding matched-rank PSD extension, however, one can indeed go beyond monotonicity, though this requires a condition on how the exact certificate $Q_k$ is chosen.
> > >
> > > First, we show that a stronger quantitative descent is already available from the current paper’s surrogate. Second, we explain what can be said in the matched-rank PSD case.
> > >
> > > **Explicit sufficient-decrease bound:**
> > >
> > > For the current surrogate, the within-step gap can be quantified exactly.
> > >
> > > From the paper, we have:
> > >
> > > $U_k(Y)=\lVert Y-B_k\rVert_F^2 + C_k + \frac{1}{2\lambda}\lVert Y-Y_k\rVert_F^2$
> > >
> > > $Y_{k+1}=(1-\alpha)Y_k+\alpha B_k$
> > >
> > > $\alpha=\frac{2\lambda}{1+2\lambda}$
> > >
> > > A completion of squares gives $U_k(Y)=U_k(Y_{k+1})+\frac{1}{\alpha}\lVert Y-Y_{k+1}\rVert_F^2$.
> > >
> > > Hence, the within-step gap is explicit: $U_k(Y_k)-U_k(Y_{k+1})=\frac{1}{\alpha}\lVert Y_{k+1}-Y_k\rVert_F^2$.
> > >
> > > Since $Y_{k+1}-Y_k=\alpha(B_k-Y_k)$, this is also $\alpha\lVert B_k-Y_k\rVert_F^2$.
> > >
> > > Since  $F\le U_k$ and $U_k(Y_k)=F(Y_k)$, we obtain:
> > >
> > > $F(Y_k)-F(Y_{k+1})=U_k(Y_k)-F(Y_{k+1})\ge U_k(Y_k)-U_k(Y_{k+1})=\frac{1}{\alpha}\lVert Y_{k+1}-Y_k\rVert_F^2=\alpha\lVert B_{k}-Y_k\rVert_F^2$
> > >
> > > Therefore, from what is already available from the current paper’s surrogate, the descent is quantitatively controlled, not only monotone.
> > >
> > > **Matched-rank PSD contraction:**
> > >
> > > Now, we assume the matched-rank PSD case in which $rank(\Sigma_\star)=r$ and the factor rank is also $r$.
> > >
> > > Write $\Sigma_\star=U_r\Lambda U_r^T$.
> > >
> > > Set $S=(\Sigma_\star)^{1/2}=U_r\Lambda^{1/2}U_r^T$ and $T=U_r\Lambda^{1/2}$.
> > >
> > > Then $\Sigma_\star=TT^T$, and $\lVert SY\rVert_{\star}=\lVert T^T Y\rVert_{\star}$.
> > >
> > > If one extends the factor objective algebraically to this PSD setting, then $\widetilde F(Y)=\lVert Y\rVert_F^2+\lVert T\rVert_F^2-2\lVert T^T Y\rVert_{\star}$.
> > >
> > > Equivalently, $\widetilde F(Y)=\min_{R \in \mathbb{R}^{r\times r},\ R^T R = I_r}\lVert Y-TR\rVert_F^2$.
> > >
> > > where $R$ is an $r\times r$ orthogonal matrix.
> > >
> > > Choose $R_k\in\arg\max_{R \in \mathbb{R}^{r\times r},\ R^T R = I_r} \operatorname{tr}(R^\top T^\top Y_k)$ so that $\widetilde F(Y_k)=\lVert Y_k-TR_k\rVert_F^2$.
> > >
> > > Here the **exact certificate** is $Q_k$: a feasible maximizer (with orthonormal columns) satisfying
> > >
> > > $tr(Q_k^T S Y_k)=\lVert SY_k\rVert_{\star}$
> > >
> > > In the matched-rank PSD case, we choose the subspace aligned certificate
> > >
> > > $Q_k=U_rR_k$.
> > >
> > > Then $tr(Q_k^T S Y_k)=tr(R_k^T T^T Y_k)=\lVert T^T Y_k\rVert_{\star}=\lVert SY_k\rVert_{\star}$.
> > >
> > > Thus $Q_k$ is an exact maximizer for the original certificate problem, and $B_k=SQ_k=TR_k$.
> > >
> > > Therefore the ITSPACE update becomes $Y_{k+1}=(1-\alpha)Y_k+\alpha TR_k$.
> > >
> > > Hence, $\widetilde F(Y_{k+1})\le \lVert Y_{k+1}-TR_k\rVert_F^2$.
> > >
> > > Since $Y_{k+1}-TR_k=(1-\alpha)(Y_k-TR_k)$,
> > >
> > > we have $\lVert Y_{k+1}-TR_k\rVert_F^2=(1-\alpha)^2\lVert Y_k-TR_k\rVert_F^2$.
> > >
> > > Because $\lVert Y_k-TR_k\rVert_F^2=\widetilde F(Y_k)$, we get
> > >
> > > $\widetilde F(Y_{k+1})\le (1-\alpha)^2\widetilde F(Y_k)$.
> > >
> > > Iterating this one-step inequality gives:
> > >
> > > $\widetilde F(Y_k)\le (1-\alpha)^{2k}\widetilde F(Y_0)$.
> > >
> > > Thus in this matched-rank PSD case, we have that $\widetilde F$ contracts linearly with factor $(1-\alpha)^2$.
> > >
> > > What prevents us from stating this as the general singular-case theorem is the non-uniqueness of exact singular certificates. Without the subspace-aligned choice of $Q_k=U_rR_k$ above, non-target fixed points can occur.
> > >
> > > **Counterexample showing why a blanket singular-case theorem fails:**
> > >
> > > Take $\Sigma_\star=diag(1,1,0)$,   $r=2$,   $Y_0=[e_1,0]$.
> > >
> > > Then $Q_0=[e_1,e_3]$
> > >
> > > is an exact maximizer, since $tr(Q_0^TSY_0)=1=\lVert SY_0\rVert_\star$,
> > >
> > > but it gives $B_0=Y_0$.
> > >
> > > Hence $Y_1=Y_0$ even though $Y_0Y_0^T\ne \Sigma_\star$.
> > >
> > > So your intuition is correct in the matched-rank PSD setting, but the certificate-selection caveat is essential. We will add both the explicit sufficient-decrease bound and this matched-rank PSD contraction result in the appendix.

---

### Official Review · Reviewer_9Gj6 · 2026-03-07

**Soundness:** 2
**Presentation:** 1
**Significance:** 2
**Originality:** 2
**Overall Recommendation:** 4
**Confidence:** 2

**Summary:**

This paper considers the optimal transport problem of aligning Gaussians: namely, consider two centered Gaussians with covariances $X$ and $\Sigma^\star$: the objective is to procedurally align $X$ to $\Sigma^\star$ via a sequence of iterates $X_1,\dots,X_k$, and progress at iterate $i$ is determined by the squared Wasserstein-2 distance between the gaussian distributions having covariances $X_i$ and $\Sigma^\star$ (which has a closed-form expression, called the Bures-Wasserstein (BW) objective)). The authors mention that "BW-faithful" methods that directly optimize the objective over PSD matrices have expensive inner-loop computations. This prompts the authors to instead consider an alternate parameterization of the iterates $X_i$ as low-rank PSD matrices, and perform updates over them. The authors propose an algorithm ITSPACE that parameterizes each $X_i$ as $Y_iY_i^T$, so that the intermediate covariances are PSD by construction. The main observation seems to be that the closed-form BW objective admits a "difference-of-convex-representation in factors", which allows the authors to derive updates based on majorization-minimization. The authors show that with exact computations, the iterates monotonically decrease the BW objective. They also consider the case where computations are inexact, and track how the errors may be controlled. Finally, the authors perform experiments to show that the proposed ITSPACE method runs fast and minimizes the BW objective at a faster rate than other benchmarks.

**Compliance With Llm Reviewing Policy:**

Affirmed.

**Final Justification:**

I am still not fully convinced of the problem setting considered in this paper. The authors' response partially addresses my confusions, but it is my view that the paper could use a rewrite explaining why this problem setting is interesting, and emphasizing more on how the authors' solution is an efficient implementation compared to the baseline. I am happy to upgrade my score from 3 to 4, but I remain not entirely convinced that this paper is an accept; therefore, my score is really a 3.5

**Key Questions For Authors:**

1) Maybe the authors believe this is obvious from the writing, or obvious to the Optimal Transport community, but the motivation of the problem was not at all clear to me. The problem setup assumes that the target covariance $\Sigma^\star$ is given as part of the input, along with an initial covariance $X_0$. If we are already given the target $\Sigma^\star$, why are we concerned with producing a step-by-step sequence $X_1,\dots,X_k$ of iterates that minimizes W2 distance to $\Sigma^\star$? I guess it is not clear to me why the naive solution in the introduction, i.e., set current covariance to target, is not what we might desire.

2) The authors say while comparing to BW-faithful methods (Lines 154-161) that these typically employ $d \times d$ matrix operations in each iteration. It was not entirely clear to me what the computational cost of these operations is, and how it compares to the per-iteration cost $O(d^2r+dr^2+r^3)$ of ITSPACE. Could the authors elaborate on this? Why is it immediate that the per-iteration cost of ITSPACE is better than that of these BW-faithful methods?

**Limitations:**

yes

**Strengths And Weaknesses:**

### Strengths

The theory shows that the proposed algorithm for constructing a sequence of covariances monotonically decreases the objective. The parameterization of the algorithm ensures that the intermediate iterates are bona fide covariances. The low-rank factorization seemingly reduces per-iteration cost, and keeps algebraic operations numerically stable. The empirical results are clearly strong: the plot in Figure 1 shows how ITSPACE decreases the objective faster per iteration than other methods, whereas the numbers in Tables 1 and 2 are also flattering.


### Weaknesses

My main issue with the paper is its writing. While I am mathematically literate, I will admit that this paper is outside my area of expertise. That said, I believe that the writing in the paper is **extremely dense**. Most written sentences are extremely hard to parse, and jam-packed with technical jargon. There seems to be no hand-holding whatsoever provided for readers who might not be entirely familiar with all the jargon. Admittedly, this need not always be the case for papers that deal with involved math: well-written papers convey the central ideas with intuitive examples, and accessible language. In this regard, I believe the paper is severely lacking, and it makes me feel like the content and results of the paper may not really be accessible and understandable to a broad audience. For this reason, I am not in favor of accepting the paper in its present form. This is not to say that the derived theoretical and empirical results are weak. But if the language of the paper is such that the results aren't really understandable, then I do not see their value. Furthermore, there also isn't much intuition provided for why people should care about this problem.

In summary, I feel like the paper could be written much more accessibly, if its ultimate objective is to disseminate knowledge and enlighten the community.

---

> ### Author Rebuttal · Authors · 2026-03-31
>
> Thank you for the helpful review. Below we address the motivation, readability, and technical questions in turn.
>
> **Answer for Weakness and Q1:**
>
> Thank you for this comment; the main motivation is the following. In applications such as domain adaptation, the goal is to align a source distribution with a target distribution, and optimal transport is one principled way to do this by minimizing the cost of moving one distribution toward the other. In many modern pipelines, however, the source and target distributions are not manipulated directly in raw sample space; they are represented in a latent/feature space, often through covariance summaries. In that setting, the task is to gradually update one covariance summary so that it gets closer to the target covariance, where closeness is measured by the Gaussian Wasserstein / Bures--Wasserstein distance.
>
> The paper studies the practically constrained version of that problem. If there were no constraints, then the trivial solution would be to set $X=\Sigma^\star$. But in the regime we study, the covariance is not a free variable that can simply be replaced once and for all. It is a state inside a larger pipeline: it is estimated from current features, updated under tight compute and memory budgets, and may be used immediately downstream after only a very small number of updates. This creates three concrete requirements: (i) each intermediate iterate must remain a valid covariance, (ii) the update must respect a low-rank budget, and (iii) only a few steps are affordable.
>
> In that regime, the relevant question is not whether one can jump to the target covariance in principle, but whether one can make meaningful progress toward it in a few cheap, valid steps under the exact Gaussian OT / BW objective.
>
> We will add more tutorial background on optimal transport and this application path in the revision.
>
> **Q2:** The key difference is the repeated inner-loop update cost.
>
> **BW-faithful full-rank methods** repeatedly apply expensive operations to a full $d\times d$ matrix at every iteration; for example, matrix square roots, inverses, logarithms, or closely related linear-algebra solves. This is why their per-step cost is summarized as $O(d^3)$.
>
> By contrast, **ITSPACE** computes the target square root $S=(\Sigma^\star)^{1/2}$ once, and then each iteration updates only a $d\times r$ factor using two dense multiplies plus a polar step on a $d\times r$ matrix, giving per-step cost $O(d^2r+dr^2+r^3)$.
>
> Thus, the advantage arises in the rank-budgeted regime studied in the paper, where $r\ll d$: ITSPACE updates a low-rank factor instead of repeating full-matrix BW operations at every step. This is the setting in which the paper makes its complexity claim.

---

> > ### Author Rebuttal · Reviewer_9Gj6 · 2026-04-04
> >
> > Thank you, Your clarification is indeed helpful.. The motivation of updating gradually now makes sense, given that we can only update iteratively under a tight budget. But maybe I am still missing something, so as a follow-up:
> >
> > It would appear that the cost of directly setting $X_0$ to the target is $\Omega(d^2)$ operations.
> >
> > From the author's response above, the cost of each iteration of ITSPACE seems to be $O(d^2r+dr^2+r^3)$.
> >
> > In this case, why should I think of the latter fitting under a "tight-budget", whereas the former doesn't?

---

> > > ### Author Response · Authors · 2026-04-06
> > >
> > > Thank you for your response. Yes, a literal overwrite $X \leftarrow \Sigma_\star$ is only an $O(d^2)$ write. However, that is the trivial unconstrained endpoint, not the repeated rank-budgeted update problem studied in the paper.
> > >
> > > In the paper's setting, the full target covariance $\Sigma_\star$ is the object we want to approach, but it is **not** the state the algorithm is allowed to keep after each step. The maintained iterate is a different object: it must always remain a valid PSD covariance state of rank at most $r$, because that is the compact state the next part of the pipeline actually uses. Thus, at each iteration, the algorithm is only allowed to keep a **a low-rank covariance state** with a budget of $r$.
> > >
> > >
> > > This matters because the current iterate may already be the one used by the next stage of the pipeline after only $1$, $2$, or $5$ updates, not just at the very end. A concrete ML example is whitening--recoloring style transfer: the covariance representation available at the current step already determines the current feature transform, and therefore the current stylized output. So the intermediate state is not just a path toward the endpoint; it is the object the next stage actually uses.
> > >
> > >
> > > For that reason, every method in our experiments must output a valid low-rank covariance state at every step. Concretely, we enforce $r=16$ uniformly, and full-rank baselines are projected back to rank $r$ after each update. Under that shared protocol, replacing the iterate by the full dense target $X \leftarrow \Sigma_\star$ is not a feasible step when $r<d$, because it violates the rank budget and leaves the low-rank state space being compared.
> > >
> > >
> > > Thus "tight-budget" here is not the cost of a single assignment; it is the cost of repeated updates that stay inside the low-rank state representation and yield a usable rank-bounded state at every step. We will make this distinction explicit.

---

### Official Review · Reviewer_RKaH · 2026-03-07

**Soundness:** 3
**Presentation:** 3
**Significance:** 2
**Originality:** 3
**Overall Recommendation:** 4
**Confidence:** 3

**Summary:**

Given a centered Gaussian probability distribution with covariance matrix $\Sigma^* \in \mathbb{R}^d$, the authors propose a new optimization algorithm (ITSPACE) to approximate $\Sigma^{\star}$ by a covariance matrix $X$ whose rank is arbitrarily limited to $r << d$. This approximation is made in order to minimize the Wasserstein 2 distance $W_2^2(\mathcal{N}(0,X), \mathcal{N}(0,\Sigma^*))$. At each iteration of the algorithm, a covariance matrix $X_k$ is calculated. To ensure that $X_k$ is PSD and that the rank constrain is satisfied, the update is done on $Y_k \in \mathbb{R}^{d \times r}$ such that $X_k = Y_k Y_k^T$. At each step ITSPACE minimises $U_k(Y)$ such that $W_2^2(YY^T,\Sigma^{\star}) \leq U_k(Y)$ and whose minimiser admits a closed form. The authors prove that at each step of the algorithm, the Wasserstein distance decreases.

**Compliance With Llm Reviewing Policy:**

Affirmed.

**Key Questions For Authors:**

Questions & remarks:
   - In the experiments how is approximated $\Sigma^{\star}$ ?
   - In the experiments the performance of the algorithm are always evaluated with BW distance which does not take into account whether the target distribution is well represented by $\mathcal{N}(0,\Sigma^*)$ and so the distribution $\mathcal{N}(0,X)$.
   - The part on the definition of polar factor in section 4 should perhaps be placed before where it is first mentioned in section 3.

**Limitations:**

yes

**Strengths And Weaknesses:**

Strenghts:
   - Well-explained consistent, and solid method..
   - Theoretical guarantees on the decreasing of the Wasserstein distance during the ITSPACE stages.
   - Computational cost less then O(d^3).

Weaknesses:
   - This method only covers cases where the target distribution is fairly simple, i.e., can be approximated by a Gaussian distribution. In particular, multimodal distributions cannot be handled here.
   - All methods are evaluated with BW distance which is biased because ITSPACE aims to minimize this distance, whereas other methods with which it is compared aim to minimize another distance.

---

> ### Author Rebuttal · Authors · 2026-03-31
>
> We appreciate the thoughtful and encouraging review. Below we address the comments and questions.
>
> **W1:** The Gaussian assumption in our approach can be made in the embedding/latent space, rather than only in raw input space. This is already common in ML, including VAEs [1], Gaussian transport formulations in generative modeling [2,3], and broader Bures–Wasserstein uses such as the graph-alignment / barycenter directions highlighted by Reviewer 4 / Q4 [4,5]. Thus, the relevant regime is broader than data that appear Gaussian in the raw input space.
> The same viewpoint is also relevant in multimodal settings, where a shared latent space is used across modalities (e.g., text and image). In that case, related multimodal generative models also model the latent variable in the standard VAE way via a Gaussian prior [6], so the Gaussian assumption is made on the latent embedding rather than on raw modalities themselves.
>
> [1] Kingma & Welling, Auto-Encoding Variational Bayes, ICLR 2014
>
> [2] Bunne et al., The Schrödinger Bridge between Gaussian Measures has a Closed Form, AISTATS 2023
>
> [3] Haviv et al., Wasserstein Flow Matching: Generative Modeling Over Families of Distributions, ICML 2025
>
> [4] Wang et al., On the Bures–Wasserstein Distance between Graph Laplacians for Graph Matching, WACV 2024
>
> [5] Chewi et al., Gradient Descent Algorithms for Bures–Wasserstein Barycenters, COLT 2021
>
> [6] Wu & Goodman, Multimodal Generative Models for Scalable Weakly-Supervised Learning, NeurIPS 2018
>
> **W2:** We do not evaluate only with BW. We evaluate (i) BW distance for covariance-level optimality, (ii) downstream predictive performance (AUROC / accuracy), and (iii) runtime.
> BW is intentional because, for centered Gaussians with covariances $X$ and $\Sigma^\star$, it is exactly $W_2^2(\mathcal N(0,X), \mathcal N(0,\Sigma^\star))$, i.e., the exact Gaussian OT objective on covariances. This lets us ask a single common question: under the same rank and step budgets, which update rule reduces the target discrepancy fastest? The direct like-for-like comparisons are therefore the BW-faithful baselines (BW geodesic and BW-GD), which optimize the same objective; Euclidean / Log-Euclidean / AIRM / Sinkhorn are included diagnostically to test whether alternative geometries or regularized OT also improve this same BW target. Evaluating each method only on its own native loss would compare different objectives and therefore would not answer the paper's question.
>
> **Downstream performance:** Experiment II thus evaluates downstream predictive performance: AUROC on Camelyon17 and classification accuracy on VisDA-2017 / Terra, after the aligned covariance is used in the same whitening/recoloring + frozen-head pipeline for all methods. In this controlled setting, ITSPACE recovers most of the drift-induced drop within very small budgets and remains competitive with the strongest baselines downstream.
>
> **Speed:** Experiment II also evaluates how quickly each method attains its downstream AUROC / accuracy gains. The reported $t_5$ measures this downstream adaptation time.
>
> **Q1:** In Experiment I, $\Sigma^\star$ is the empirical centered covariance of the designated target-domain features (App. E.1), i.e., $\hat\Sigma=\frac{1}{n-1}(Z-\bar Z)^\top(Z-\bar Z)$, followed by symmetrization. If a baseline requires SPD input, we apply the same small diagonal floor described in App. E.1. In CovDrift-MR, the target moments $(\mu_t,\Sigma_t)$ are likewise estimated empirically from the drifted unlabeled target features (App. F.1).
>
> Thus, the target covariance is estimated from data rather than assumed known in closed form, with the same moment-estimation / stabilization protocol across methods.
>
> **Q2:** As noted in **W2**, the paper does not evaluate performance only with BW distance; it also reports downstream AUROC / accuracy and runtime.
>
> More importantly, in the setting studied here, BW is exactly the metric that arises when the target and current distributions are represented by $\mathcal N(0,\Sigma^\star)$ and $\mathcal N(0,X)$ [1,2,3].
>
> The introduction states that, under a centered Gaussian approximation, the squared Wasserstein-2 discrepancy reduces to the BW objective on covariances, and Sec. 2.2 gives this closed form explicitly. Hence, in the setting studied here, BW is the exact Gaussian OT objective, not an ad hoc proxy.
>
> As discussed in **W1**, Experiment II is then included to evaluate downstream task performance, rather than only covariance-level BW distance.
>
> [1] Gelbrich, On a Formula for the $L_2$​-Wasserstein Metric Between Measures on Euclidean and Hilbert Spaces, Math. Nachr., 1990
>
> [2] Takatsu, On the Wasserstein Geometry of Gaussian Measures, Probab. Theory Relat. Fields, 2011
>
> [3] Bhatia, Jain & Lim, On the Bures–Wasserstein Distance Between Positive Definite Matrices, Expositiones Mathematicae, 2019
>
> **Q3:** The polar factor is already defined in Sec. 3.2 immediately before Eq. (8), including the thin-SVD form $UV^\top$.

---

> > ### Author Rebuttal · Reviewer_RKaH · 2026-04-03
> >
> > I would like to thank the authors for their response and clarification. I am keeping my rating.

---

> > > ### Author Response · Authors · 2026-04-06
> > >
> > > Thank you for confirming that your concerns were fully resolved. We are glad the clarifications were helpful.

---

### Official Review · Reviewer_mh5s · 2026-03-12

**Soundness:** 3
**Presentation:** 3
**Significance:** 2
**Originality:** 3
**Overall Recommendation:** 3
**Confidence:** 2

**Summary:**

This paper provides an algorithm for computing a trajectory over covariances that reduces the Wasserstein distance between two centered Gaussian distributions. The authors use proximal majorization-minimization on the Bures-Wasserstein closed form for the Wasserstein distance between two Gaussians. The algorithm provides iterates that are constrained with a rank-budget, and preserve the PSD structure of the covariance matrix. The authors study certain theoretical properties of the algorithm, and provide experiments that show the benefit of this algorithm for covariance adaptation with few steps.

**Compliance With Llm Reviewing Policy:**

Affirmed.

**Final Justification:**

Given that a comparison with OT displacement was included, I'll increase the score to 3. However, I think that to recommend acceptance the paper would either need to be fundamentally stronger on the theory side (e.g. Q2 of Reviewer tLWT could be a good next step), or fundamentally stronger on the experiments with ablations that show how the ITSPACE iterates are mechanistically different from the iterates of other algorithms.

**Key Questions For Authors:**

* When reporting times in the main text, I think it would be very helpful to describe the hardware and computing setup, and the extent of parallelization that can be used.
* For algorithms that do not have rank constraints/projections themselves, how much of the algorithm time is spent on projection?

*Minor comments*:

* Based on the definition of polar, I think line 191 has a mistake, we must have $\operatorname{polar}(A) = \Sigma V^\top$.
* Figure 2 axes titles and legend could use a larger font size.

**Limitations:**

I think the authors can better highlight the fact that the interpolation they are following is only principled under a Gaussian assumption, which can be restrictive in practice.

**Strengths And Weaknesses:**

**Strengths**: The algorithm has a clean mathematical derivation, and Figure 1 demonstrates that under rank constraints, it can align the covariance with the target remarkably quickly.

**Weakness**: I'm not familiar with the literature on interpolating two covariance matrices using different geometries. However, to me it seems like neither the experimental gains nor the theory are rich enough:
* For example, the tradeoff between choosing a low rank-budget and final downstream performance is not clear.
* The authors do not consider simple baselines such as using the OT displacement itself (which would involve a one-time inversion of matrices), and the efficiency/utility tradeoff between using exact OT displacements vs. ITSPACE is not clear.
* In terms of downstream performance, seems like there is no difference between ITSPACE and BW-GD in Table 2.
* Further, some algorithms may perform efficiently without rank projections, e.g. Euclidean interpolation. In that case, additionally performing low-rank projections as done in the experiments here might artificially reduce their accuracy on downstream tasks.

---

> ### Author Rebuttal · Authors · 2026-03-31
>
> Thank you for the feedback. Below we address the comments.
>
> W1: The paper is not aiming at a full downstream-versus-rank tradeoff. It studies few-step adaptation under a fixed low-rank budget. This is the relevant regime when covariances are maintained under tight compute/memory constraints (e.g., in edge devices) and consumed immediately downstream (e.g., in domain adaptation or whitening/normalization pipelines). The present claim is therefore a fixed-budget comparison, not an all-rank scaling claim.
>
> W2: The exact OT displacement baseline is already included in the paper: the BW geodesic (i.e., BW-faithful closed-form Gaussian displacement interpolation) baseline.
>
> The comparison is thus already visible in the results: ITSPACE gives better few-step utility than BW geodesic under the same rank-budgeted setting. For example, at $K=2$, ITSPACE exceeds BW geodesic on Camelyon17 (74.01 vs. 72.86), VisDA-2017 (84.26 vs. 78.14), and Terra (63.94 vs. 60.44). In the rank-budgeted alignment experiment, ITSPACE also reaches the BW-gap thresholds much earlier, while BW geodesic often does not reach the tighter threshold within $K=20$.
>
> Thus, the paper already includes the exact-displacement baseline and shows the practical tradeoff in the few-step rank-budgeted regime.
>
> W3: Table 2 shows that ITSPACE and BW-GD are close on Camelyon17, but that is not the full picture.
>
> At small budgets, the broader downstream results favor ITSPACE. Appendix Table 6 shows a clearer advantage on VisDA-2017 (84.26 vs. 82.82 at $K=2$) and a smaller but still positive advantage on Terra (63.94 vs. 63.56 at $K=2$).
> More importantly, in the rank-budgeted alignment benchmark, ITSPACE reaches the BW-gap targets far faster than BW-GD, with about $198\times$ to $320\times$ speedups across datasets.
>
> Therefore, ITSPACE achieves better or comparable downstream performance, whilst being much more effective in the strict few-step rank-budgeted setting studied in the paper.
>
> W4: Projection is not an extra handicap; it is part of the problem definition studied here. The paper explicitly targets a regime where each step must output a rank-$r$ covariance/factor that can be consumed immediately downstream. For naturally full-rank methods, projection is simply the cost of enforcing that shared requirement. In CovDrift-MR, all methods share the same standardization, moment estimates, rank enforcement, and transport-map implementation; only the covariance-alignment update differs.
>
> Unconstrained full-rank comparisons correspond to different and more expensive problems, in time and memory, than the rank-budgeted problem studied here. Several full-rank baselines still reach a similar final performance by $K=20$, hence the issue is few-step utility under the rank budget, rather than an artificial loss of eventual accuracy.
>
> Q1: We will add this. Runtimes were measured using one GPU per run (NVIDIA RTX A6000; AMD EPYC 7763; about 503 GiB RAM). Runs were single-device and sequential across methods/seeds, not multi-GPU or distributed. Thus, the numbers are synchronized one-GPU wall-clock times, not parallel throughput.
>
> Q2: As explained in W4, projection is not an extra handicap; it is part of the problem definition studied here. The paper evaluates a rank-budgeted setting in which every method must output a rank-$r$ covariance/factor at each step. Accordingly, the reported runtime includes the cost of enforcing that shared rank-$r$ requirement. Quantitatively, projection accounts for a substantial share of algorithm time across the relevant baselines (median about $67\%$ across methods/datasets/seeds). These projections are part of the evaluated rank-budgeted setting, not a separate auxiliary procedure.
>
> Q3: This is not a mistake: for $A = U\Sigma V^\top$, $\operatorname{polar}(A) = UV^\top$, while $(A^\top A)^{1/2} = V\Sigma V^\top$.
>
> Q4: We will enlarge the Figure 2 axis labels and legend.
>
> L: The Gaussian assumption is widely used in modern ML. In many such formulations, the latent embedding is assumed to be Gaussian rather than the raw input, for example in variational autoencoders [1] and Gaussian transport formulations [2]. In optimal transport specifically, the Bures–Wasserstein / Gaussian-$W_2$ geometry is the natural geometry in this setting [2,3].
>
> Hence the paper is working in a regime that is well-motivated in ML. As noted in our response to Reviewer 2 / W1, this setting is natural when the Gaussian assumption is made in latent/embedding space rather than in raw input space. As highlighted by Reviewer 4 / Q4, related Bures–Wasserstein formulations are already being considered for structured uses such as barycenters.
>
> We will revise the paper to make this motivation and scope more explicit.
>
> [1] Kingma & Welling, Auto-Encoding Variational Bayes, ICLR 2014
>
> [2] Bunne et al., The Schrödinger Bridge between Gaussian Measures has a Closed Form, AISTATS 2023
>
> [3] Chewi et al., Gradient Descent Algorithms for Bures–Wasserstein Barycenters, COLT 2021

---

> > ### Author Rebuttal · Reviewer_mh5s · 2026-04-03
> >
> > I thank the authors for their response. Given that a comparison with OT displacement was included, I'll increase the score to 3. However, I think that to recommend acceptance the paper would either need to be fundamentally stronger on the theory side (e.g. Q2 of Reviewer tLWT could be a good next step), or fundamentally stronger on the experiments with ablations that show how the ITSPACE iterates are mechanistically different from the iterates of other algorithms.

---

> > > ### Author Response · Authors · 2026-04-06
> > >
> > > Thank you for the score update and for clarifying the remaining concerns. We agree that one strong addition is a clearer theoretical answer to the point you highlighted via Reviewer tLWT's Q2.
> > >
> > > On that point, we can sharpen the paper beyond monotonicity. The MM surrogate yields the explicit sufficient-decrease bound $F(Y_k)-F(Y_{k+1}) \ge \frac{1}{\alpha}\lVert Y_{k+1}-Y_k\rVert_F^2$.
> > >
> > > Since $Y_{k+1}-Y_k=\alpha(B_k-Y_k)$, this is equivalently $F(Y_k)-F(Y_{k+1}) \ge \alpha \lVert B_k-Y_k\rVert_F^2$.
> > >
> > > More importantly, in the matched-rank PSD special case with $rank(\Sigma_\star)=r$, and under a natural subspace-aligned exact-certificate selection, one obtains a linear contraction result in that regime.
> > >
> > > In our recent response to Reviewer tLWT, we explained this sharpened analysis, together with a concrete counterexample showing why a blanket singular-case theorem is not valid without specifying how the exact singular certificate is chosen.
> > >
> > > On the empirical side, note that **Figure 2** already shows a more effective early trajectory under the shared rank budget. The BW-gap curves separate early under the common $r=16$ budget, and **Table 5** shows ITSPACE reaching the BW thresholds about $198\times$ to $320\times$ faster than BW-GD across datasets. The same early-trajectory effect appears downstream at small budgets. At $K=2$, ITSPACE is above BW geodesic on Camelyon17, VisDA-2017, and Terra. Versus BW-GD, it is higher on VisDA-2017, slightly above on Terra, and essentially tied on Camelyon17.
> > >
> > > Taken together, the follow-up discussion sharpens the theory in exactly the direction you pointed to, and the current experiments already show that ITSPACE follows a more effective early trajectory in the strict rank-budgeted regime studied here. We hope this addresses your concerns.

---

### Decision · Program_Chairs · 2026-04-30

**Decision:**

Accept (regular)

**Comment:**

The authors propose a low-rank iterative method (ITSPACE) for aligning covariance matrices under the Bures–Wasserstein (BW) distance. The proposed approach leverages a majorization–minimization framework with closed-form updates in a factorized representation to exploit the low-rank setup, and ensure its PSD. We think the proposed approach is technically solid, efficiently exploit low-rank setup in computation. However, the reviewers also raised concerns on its limited experimental depth with modest downstream improvements and a restricted scope to Gaussian/covariance-based settings. There are mixed opinions from the reviewers on the submission. Overall, there are concerns on the restricted scope, and its empirical evidence, we think the proposed approach is efficient, and it is interesting for a subcommunity involving data analysis with PSD/Gaussian/covariance structure.